# scRNA-seq of gastric tumor shows complex intercellular interaction with an alternative T cell exhaustion trajectory

Keyong Sun[1,7], Runda Xu[1,7], Fuhai Ma[2,3,7], Naixue Yang[1,4,7], Yang Li[2,7], Xiaofeng Sun[1,5], Peng Jin[2], Wenzhe Kang[2], Lemei Jia[1], Jianping Xiong[2], Haitao Hu[2], Yantao Tian[2]✉ & Xun Lan[1,4,5,6]✉

The tumor microenvironment (TME) in gastric cancer (GC) has been shown to be important for tumor control but the specific characteristics for GC are not fully appreciated. We generated an atlas of 166,533 cells from 10 GC patients with matched paratumor tissues and blood. Our results show tumor-associated stromal cells (TASCs) have upregulated activity of Wnt signaling and angiogenesis, and are negatively correlated with survival. Tumor-associated macrophages and *LAMP3*+ DCs are involved in mediating T cell activity and form intercellular interaction hubs with TASCs. Clonotype and trajectory analysis demonstrates that Tc17 (*IL-17*+*CD8*+ T cells) originate from tissue-resident memory T cells and can subsequently differentiate into exhausted T cells, suggesting an alternative pathway for T cell exhaustion. Our results indicate that *IL17*+ cells may promote tumor progression through *IL17*, *IL22*, and *IL26* signaling, highlighting the possibility of targeting *IL17*+ cells and associated signaling pathways as a therapeutic strategy to treat GC.

Gastric cancer (GC), comprising many molecular subtypes, is the fifth most common malignancy worldwide, but the third leading cause of cancer-related mortality, with an estimated 783,000 deaths in 2018[1,2]. While GC is highly treatable at the early primary stage, most patients are detected at the advanced or metastatic stage with a relatively poor prognosis. Immunotherapy, especially antibodies targeting PD-1 and CTLA4, has caused a paradigm shift for the treatment of various cancer types, such as melanoma, but the response rate in GC is relatively low[3]. Many previous studies suggested that the intertumoral heterogeneity and individual variation of cellular composition were associated with survival[4], highlighting an unmet need to dissect the complex and dynamic biological characteristics of the tumor microenvironment (TME) to exploit advanced interventions to combat it.

Recently, single-cell RNA-sequencing (scRNA-seq) has been successfully used to decipher the ecosystems of GC, to dissect and discover the underlying tumor biology of interest[5-10]. For example, Wang et al. and Zhang et al., indicated the transcriptional heterogeneity and lineage diversity in primary and metastatic gastric adenocarcinoma, and provided signature genes for diagnosis and prognosis[6,7]. Zhang et al. and Yin et al., delineated the vast cellular phenotypic remodeling during GC occurrence and development, and also identified makers for early GC detection[5,9]. Kumar et al. showed an increased plasma cell proportions in diffuse-type gastric tumors and studied the INHBA-FAP axis in cancer-associated fibroblasts[11].

In this work, we apply scRNA-seq to map the transcriptional landscape of immune, stromal, and epithelial compartments in

[1]School of Medicine, Tsinghua University, 100084 Beijing, China. [2]Department of Pancreatic and Gastric Surgery, National Cancer Center, National Clinical Research Center for Cancer, Cancer Hospital, Chinese Academy of Medical Sciences and Peking Union Medical College, No. 17 Panjiayuan Nanli, 100021 Beijing, China. [3]Department of General Surgery, Department of Gastrointestinal Surgery, Beijing Hospital, National Center of Gerontology, Institute of Geriatric Medicine, Chinese Academy of Medical Sciences, 100730 Beijing, China. [4]Peking-Tsinghua-NIBS Joint Graduate Program, Tsinghua University, 100084 Beijing, China. [5]Centre for Life Sciences, Tsinghua University, 100084 Beijing, China. [6]MOE Key Laboratory of Bioinformatics, Tsinghua University, 100084 Beijing, China. [7]These authors contributed equally: Keyong Sun, Runda Xu, Fuhai Ma, Naixue Yang, Yang Li. ✉e-mail: tianyantao@cicams.ac.cn; xlan@mail.tsinghua.edu.cn

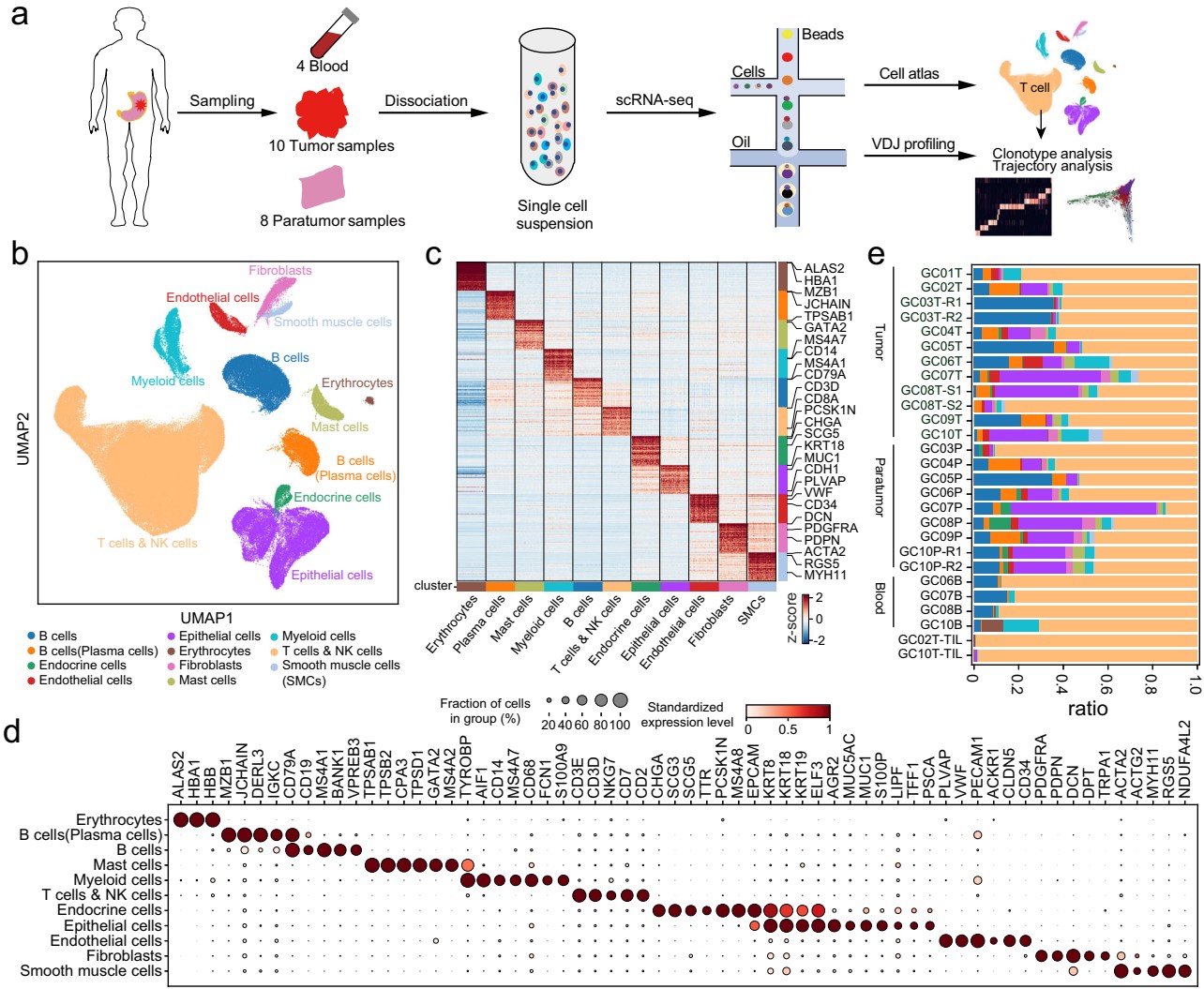

**Fig. 1 | Characterization of the gastric cancer tumor microenvironment by scRNA-seq. a** Workflow depicting sample processing and scRNA sequencing of gastric cancer (GC), peripheral blood, and paratumor cells and subsequent analytical methods. **b** Uniform Manifold Approximation and Projection (UMAP) of 166,533 single cells from 10 patients, colored by major cell types. **c** Heatmap showing the differentially expressed genes (rows) across major cell types (columns), with canonical marker genes indicated. **d** Dot plots showing marker genes for clusters in Fig. 1b. Dot size indicates the proportion of expressing cells, colored by standardized expression levels. **e** Fractions of cell types detected in each sample, colored as in Fig. 1b. GC01–GC10 represent 10 GC patients; B/P/T represent cells isolated from blood, paratumor, and tumor tissues, respectively; TIL represents tumor-infiltrating lymphocyte; GC03T-R1/R2 and GC10P-R1/R2 represent two technical replicates; GC08T-S1/S2 represent two different sites of the same tumor tissue. Source data are provided as a Source Data file.

tumors, adjacent normal tissues, and matched peripheral blood from 10 GC patients, coupled with T/B cell receptor (TCR/BCR) repertoire profiling. Our results indicate that the stromal cells in the tumor tissue undergo a significant transformation and exhibit extensive tumor-promoting features. Cell-cell communication analysis shows that TASCs, TAMs, and *LAMP3*+DCs are important mediators of complex cellular interaction in the TME. Coupled with TCR clonotype information and trajectory analysis, we show that Tc17 in GC possibly originate from the tissue-resident memory population and can subsequently differentiate into an exhausted state, which suggests an alternative pathway for T cell exhaustion. Our results indicate that IL17+ cells and pathways mediating IL17+ cells communication with tumor cells are potential therapeutic targets for treating IL17+ positive gastric cancer.

## Results

### A single-cell RNA-seq atlas of gastric cancer microenvironment

To explore the cell type diversity that participates in gastric cancer (GC) at single-cell resolution, we generated scRNA-seq profiles of all viable cells isolated from the tumor tissues of ten primary GC patients without treatment before sampling, as well as from matched peripheral blood and adjacent normal tissue (Fig. 1a and Supplementary Fig. 1a, b and Supplementary Data 1). Meanwhile, we also performed bulk whole-exome sequencing (WES) and bulk RNA-sequencing for the same samples with a few exceptions.

We merged expression profiles across all tissues and patients and retained 166,533 cells after quality control (Fig. 1b), resulting in a comprehensive atlas encompassing the entire GC ecosystem. 48.3%, 37.2%, and 14.5% of these cells originated from tumor, paratumor, and blood tissues, respectively (Supplementary Fig. 1d). The scRNA-seq profiles were partitioned into 12 broad lineages of the immune, stromal, and epithelial compartments (Fig. 1b–d), including T cells, NK cells, B cells, plasma cells, myeloid cells, mast cells, erythrocytes, endothelial cells, fibroblast cells, smooth muscle cells (SMCs), epithelial cells, and endocrine cells. Nearly every type of cell was found in all patients, with the exception of endocrine cells, the majority of which originated from patients GC07, GC08 and GC10 (Fig. 1e and Supplementary Fig. 1e). Coupled with the scRNA-seq profiling, we

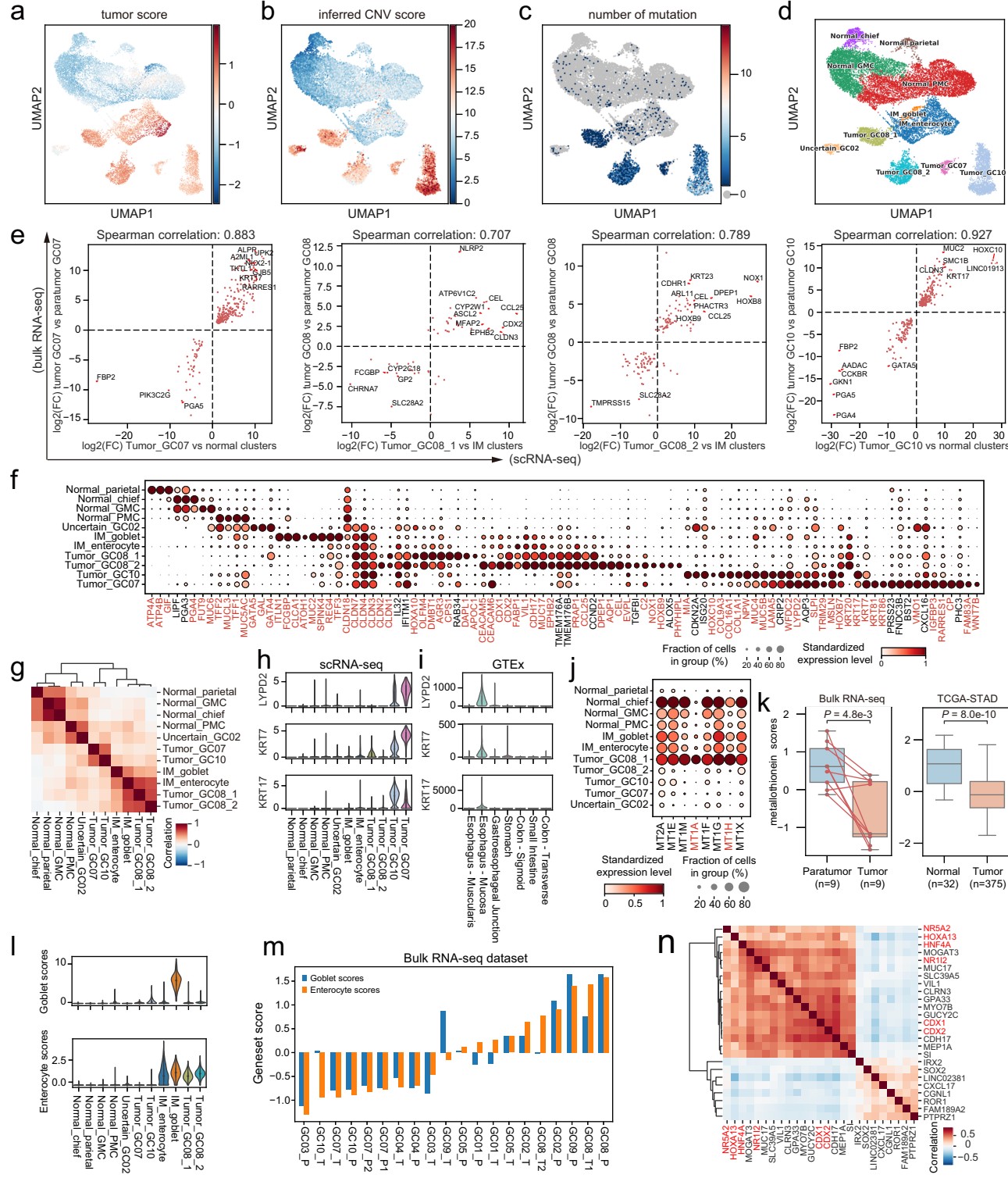

generated paired TCR and BCR (T/B cell receptor) sequencing data to investigate the state transitions within different T/B cell subtypes, as discussed in later sections (Supplementary Fig. 1f).

## Malignant cells in GC exhibit extensive heterogeneity

The cells defined as epithelial cells were extracted and re-clustered. To distinguish tumor cells and normal cells, we firstly calculated tumor scores based on the expression of signature genes of tumor and normal tissue using method described by Zhang et al.[7] (Fig. 2a). Secondly, we calculated copy number variants (CNV) scores of cells by

inferCNV[12], an algorithm to estimate the copy number changes in the genome with scRNA-seq data (Fig. 2b and Supplementary Fig. 2b). Finally, we called tumor-specific mutations for each patient by comparing the WES data of tumor tissue versus that of paratumor tissue and then, we searched the tumor-specific mutations in the matching single-cell data. Such mutations were enriched in 4 clusters of epithelial cells (Fig. 2c, Supplementary Fig. 2c, see Methods for details).

According to these results, as well as the expression patterns of cell type specific genes (Fig. 2f and Supplementary Table 1), we finally defined four main cluster types, including normal clusters, tumor

**Fig. 2 | Profiling the epithelial cells in gastric cancer at single-cell level.** UMAP of epithelial cells, colored by: **a** tumor score; **b** inferred CNV score; **c** the number of mutations. **d** UMAP of epithelial cells. Clusters are labeled with inferred cell types. GMC basal gland mucous cell, PMC pit mucous cell, IM intestinal metaplasia. **e** Scatter plot showing high correlations between the $\log_2$ fold change of DEGs from bulk RNA-seq (tumor vs paratumor) and scRNA-seq (tumor cluster vs normal/IM cluster). **f** Dot plot of marker genes for epithelial cell clusters. Dot size indicates the proportion of expressing cells, colored by standardized expression levels. Genes in red color indicated low cellular detection rates outside epithelial cells. **g** Heatmap showing the Pearson correlation between epithelial cell clusters. Violin plot showing the expression of *LYPD2*, *KRT7* and *KRT17* in the scRNA-seq dataset (**h**) and gastrointestinal tract samples from GTEx dataset (**i**). **j** Dot plot showing expression levels of metallothionein-related

genes. Dot size indicates the proportion of expressing cells, colored by standardized expression levels. Genes in red color indicated low cellular detection rates outside epithelial cells. **k** Boxplot showing metallothionein score in bulk RNA-seq dataset (left; two-sided paired *t*-test) and TCGA-STAD dataset (right; two-sided Wilcoxon rank-sum test). For all boxplots in this paper: box, interquartile range (IQR); horizontal line, median; whiskers, most extreme values within ±1.5 × IQR. **l** Violin plot showing the goblet scores (top) and enterocyte scores (bottom). **m** Bar plot showing the goblet and enterocyte scores in the bulk RNA-seq dataset. **n** Heatmap showing the combined correlation of *CDX2*-associated genes by taking the product of correlation coefficients generated from our scRNA-seq, bulk RNA-seq, and the bulk RNA-seq from TCGA-STAD. Genes in red were predicted upstream regulators of *CDX2*. Source data are provided as a Source Data file.

clusters, intestinal metaplasia (IM) clusters, and an uncertain cluster (Fig. 2d). IM_enterocyte and IM_goblet were considered as precancerous clusters as they showed neither high CNV scores nor enrichment of tumor-specific mutations. The tissue enrichment level of each cluster of cells agreed with this definition (Supplementary Fig. 2d). Besides, we found a small bunch of cells closed to goblet cells expressed several marker genes of tuft cells (Supplementary Fig. 2a). The cell number was so small that these cells could not be isolated and thus clustered into IM_enterocyte.

Each normal cluster contained cells from multiple patients, suggesting the batch effect did not affect the clustering significantly, whereas each tumor cluster was made of cells from a single patient, displaying high level of intertumoral heterogeneity (Supplementary Fig. 13f). Tumor cells from GC08 formed two distinct tumor clusters, which implied intratumoral heterogeneity. For some patients, we got very few epithelial cells. We note that only a small number of epithelial cells were detected in several patients. A possible reason was that gastric epithelial cells were more vulnerable than other cell types in our experiment, as we observed much higher mitochondrial gene percentages in epithelial cells (Supplementary Fig. 14f).

To validate the tumor cell clusters identified above, we examined if similar gene expression patterns can be observed in the bulk RNA-seq data from the same patient. Because each tumor cluster consisted of cells from a single patient, we identified the differentially expressed genes (DEGs) from scRNA-seq for each patient by comparing the tumor cell cluster vs. normal cell clusters. Similarly, we identified DEGs from bulk RNA-seq for the same patient by comparing the tumor tissue vs. the paratumor tissue. We took the intersection of the two sets of DEGs and filtered them by *p*-value, log fold change and cellular detection rate outside the epithelial cluster (see Methods). The remaining DEGs showed high correlations in expression between scRNA-seq and bulk RNA seq data (Fig. 2e), indicating that the tumor cell clusters identified using the scRNA-seq data exhibited gene expression pattern close to that of the bulk tumor tissue.

Next, we investigated the gene expression patterns of the tumor and precancerous clusters and found a widely altered expressions for claudin genes, which were responsible for tight junctions. *CLDN3*, *CLDN4* and *CLDN7* were expressed at an abnormally high level in tumor cells and IM cells (Fig. 2f). These genes were commonly expressed in intestines and esophagus mucosa, according to Genotype-Tissue Expression (GTEx) data[13] (Supplementary Fig. 2e). On the contrary, *CLDN18* was expressed by normal gastric cells and downregulated in tumor cells. These observations may due to irregular cell differentiations that are common during the development of gastric cancer.

Correlation of the mean expression of highly variable genes showed that Tumor_GC07 and Tumor GC10 were close to each other (Fig. 2g). They shared some tumor-specific DEGs commonly expressed by esophagus mucosa according to GTEx data (Fig. 2h, i). This observation might be related to the tumor site − Tumor_GC07 and

Tumor_GC10 were both from the cardias (Supplementary Data 1). However, these genes were not expressed by gastroesophageal junctions in GTEx data. DEGs between Tumor_GC08_1 and Tumor_GC08_2 showed a widely down-regulation of metallothionein genes in the latter (Supplementary Fig. 2f). These genes were expressed by normal cells and downregulated in tumor clusters except Tumor_GC08_1 (Fig. 2j, k). Meanwhile, Tumor_GC08_1 showed lower enrichment of tumor-specific mutations than Tumor_GC08_2 (Supplementary Fig. 2b). Our result suggested that Tumor_GC08_1 was likely at a less advanced stage than Tumor_GC08_2 and other tumor clusters, highlighting intratumoral heterogeneity.

## Identifying potential regulatory factors driving intestinal metaplasia

To better evaluate the IM levels of the patients, we calculated goblet scores and enterocyte scores for scRNA clusters and bulk data (Fig. 2l, m) using cell type signature genes (Supplementary Table 1). GC08 and GC09 showed high levels of IM in both tumor samples and paratumor samples. Although the pathological classification of tumors from GC07 and GC10 were intestinal type and mixed type, respectively (Supplementary Data 1), both tumors did not show obvious expression of intestinal genes.

*CDX2* is considered as the master transcription factor for IM[14,15]. To search genes associated with *CDX2* in a robust manner, we combined our two datasets as well as the Stomach Adenocarcinoma (STAD) dataset of The Cancer Genome Atlas (TCGA)[16] to find genes that are correlated with *CDX2* in expression. The Spearman's rank correlation coefficients from three datasets were multiplied together and the genes were filtered by a low cellular detection rate outside epithelial cells in scRNA-seq to avoid confounding effect from other cell types (see Methods). Genes highly correlated with *CDX2* included known *CDX2* targets, such as, *GUCY2C*, *CDH17*, *SI* and *GPA33*[17,18] (Fig. 2n). *CDX1* was another homeobox gene expressed in distal intestine and could induce IM[19,20]; *SOX2* was reported to be responsible for gastric specification and could interact with *CDX2* at protein level[21,22].

In addition to providing potential *CDX2* targets, we also examined upstream regulatory TFs of *CDX2* (shown in red in Fig. 2n) in epithelial cells by applying SCENIC, which predicts downstream targets of TFs based on gene expression and enrichment of TF motifs[23]. *HNF4A* was previously reported to regulate *CDX2* in the presence of *GATA6*, *TCF4* and β-catenin[24], while the function of *HOXA13*, *NR5A2* and *NR1I2* in the regulation of *CDX2* needed further study. We conducted overexpression experiments with gastric cancer cell lines and found that *HOXA13* could upregulate *CDX2* in SGC-7901 cells but not in MKN-28 cells (Supplementary Fig. 3b, c). Meanwhile, the overexpression of *CDX2* showed varying degrees of upregulation of *HNF4A*, *HOXA13*, *NR5A2*, *NR1I2* and *CDX1* and elongated the shape of cells (Supplementary Fig. 2d−f). According to these results, *HOXA13* and *CDX2* might form a positive feedback loop under certain circumstances. The underlying molecular mechanism deserves further study.

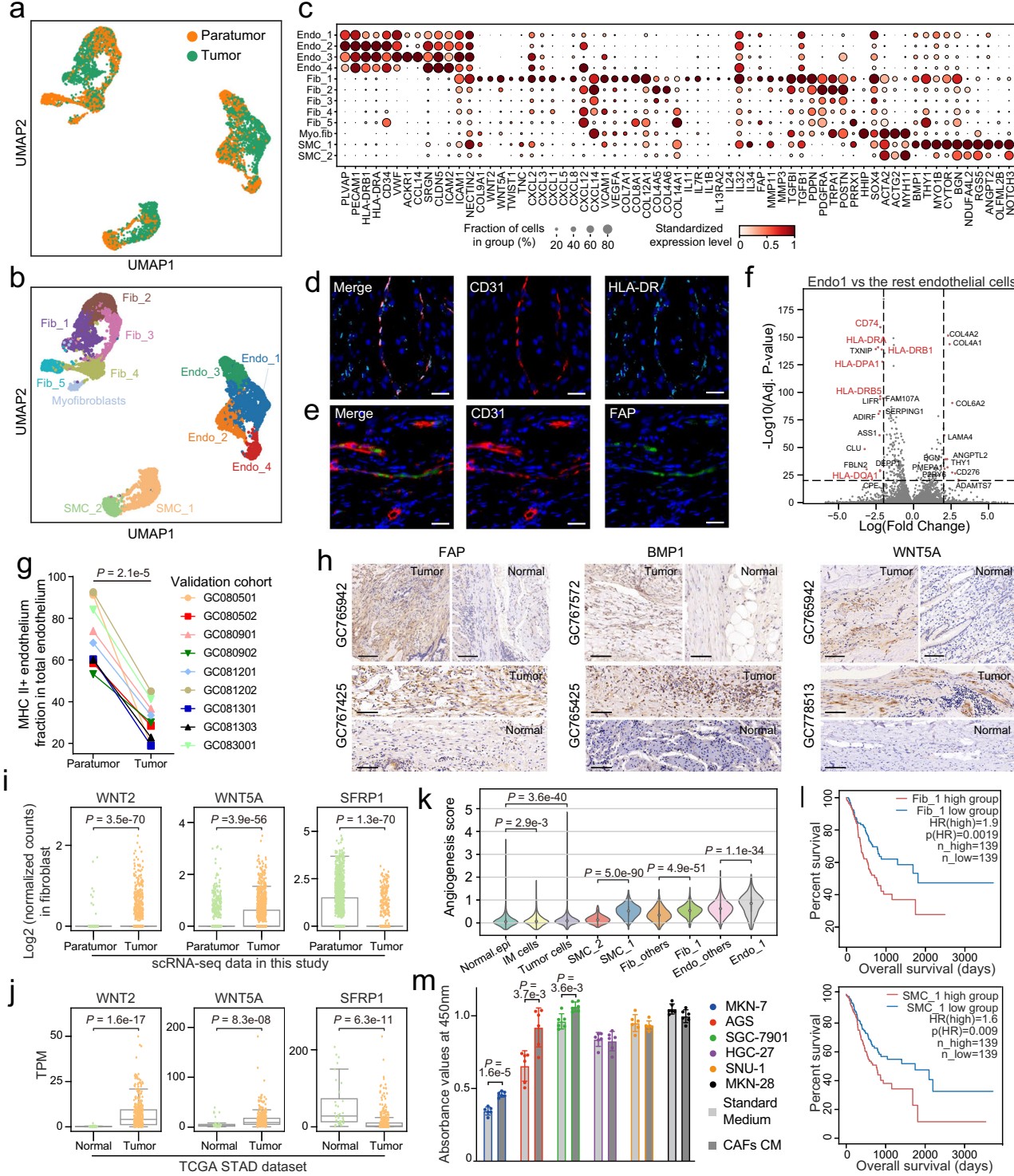

## The stromal compartment underwent substantial remodeling in GC

To decipher the function of stromal cells in the TME of GC, we re-clustered all stromal cells and found a clear separation between paratumor and tumor tissue, indicating that stromal cells in the TME have undergone global transcriptomic changes from those in paratumor tissue (Fig. 3a). We then grouped these stromal cells into 12 distinct clusters, of which Endo_1, Fib_1, and SMC_1 were predominantly enriched in tumor tissues (Fig. 3b and Supplementary Fig. 4a). Interestingly, stromal cells in paratumor had greater heterogeneity than those from tumors (Supplementary Fig. 4b), implying

stromal cells likely have more diverse functions under normal physiological conditions but become specialized within the TME.

Notably, we found endothelial cells in stomach expressed major histocompatibility complex (MHC) class II genes such as *HLA-DRA* and *HLA-DRB5* (Fig. 3c), which was confirmed by multicolor immunohistochemistry (IHC) staining on tumor sections from GC patients (Fig. 3d). Moreover, we found that Endo_1 featured downregulated MHC class II genes (Fig. 3f), indicating that the intrinsic antigen presentation function of Endo_1 was limited. We further performed flow cytometry on nine additional GC patient samples, which showed the fraction of MHC class II+ endothelium in paratumors was higher than

**Fig. 3 | Dynamic restructuring of stromal cells in GC.** UMAP of stromal cells colored by cellular tissue origin (**a**) and inferred cell types (**b**). **c** Dot plots showing marker genes across stromal cell subsets. Dot size indicates the proportion of expressing cells, colored by standardized expression levels. **d**–**e** Multicolor IHC staining with anti-CD31 and anti-HLA-DR antibodies showing HLA-DR+ endothelial cells (d; *n* = 6), and with anti-CD31 and anti-FAP antibodies showing FAP+ endothelial cells (e; *n* = 6). The scale bar represents 20 μm. **f** Volcano plot showing differentially expressed genes of tumor−enriched Endo_1 versus other endothelial cell types. Dotted lines indicate *p*-value < 1e−20 and |log$_2$ (FC)| > 2 (two-sided Wilcoxon rank-sum test with Bonferroni correction). **g** Dot plot showing the higher proportion of MHC class II+ endothelial cells in paratumors than that in tumors (*n* = 9) (two-sided paired *t*-test). **h** IHC staining of FAP, BMP1 and WNT5A on formalin-fixed and paraffin-embedded slides of independent biospecimens (*n* = 6). The scale bar represents 100 μm. **i** Boxplot showing the expression of *WNT*-related genes in fibroblasts from tumor (*n* = 1274) and those from paratumor (*n* = 1461). Each dot represents a single cell (two-sided Wilcoxon rank-sum test). **j** Boxplot showing the expression of *WNT*-related genes in tumor (*n* = 375) and in normal tissue (*n* = 32). Each dot represents a single sample (two-sided Wilcoxon rank-sum test). **k** Violin plot showing the angiogenesis score in stromal and epithelial subsets. Normal.epi and IM cells represent normal epithelial cells and intestinal metaplasia cells, respectively (two-sided Wilcoxon rank-sum test). **l** Kaplan−Meier curves of overall survival by stratifying the patients by high (top 40%) and low (bottom 40%) proportion of the respective cell type. High fractions of Fib_1 and SMC_1 are associated with poor prognosis in the TCGA-STAD cohort. HR (hazard ratio) and p(HR) was calculated by a Cox's proportional hazard model. **m** The supernatants from GC CAFs were collected by centrifugation as conditioned medium (CM), and incubated with six GC cell lines for 60 h. Cell survival was determined by CCK-8 assay (*n* = 6). Data are presented as mean values ± SD (two-sided *t* test). Source data are provided as a Source Data file.

that in tumors (*p* < 0.001, Student's paired *t* test) (Fig. 3g and Supplementary Fig. 4c), consistent with the observation in scRNA-seq data.

Both Endo_1 and Fib_1 expressed fibroblast activation protein (*FAP*), a classical cancer-associated fibroblast (CAF) marker. Similarly, we performed multicolor IHC staining to validate the presence of FAP+ fibroblast and endothelial cells in tumors (Fig. 3e and Supplementary Fig. 4d). Fib_1 also expressed others CAF markers, such as *MMP3* and *MMP11*, and inflammation-associated fibroblast markers (*IL11*, *IL24*) that promote carcinogenesis[25] (Fig. 3c and Supplementary Fig. 4g). Genes in the Wnt signaling pathway such as, *WNT2* and *WNT5A*, were upregulated in tumor fibroblasts while *SFRP1*, an inhibitor of Wnt signaling was downregulated (Fig. 3i). These genes also showed similar expression patterns in TCGA-STAD dataset (Fig. 3j). Note that these three genes were expressed almost exclusively by fibroblasts (Supplementary Fig. 4e).

Besides, Fib_1 cells exhibited upregulation of the *TWIST1-PRRX1-TNC* positive feedback pathway, which is known to promote the activation and expansion of CAFs in the TME[26]. Meanwhile, bone morphogenetic protein 1 (*BMP1*) and *ANGPT2*, which respectively facilitate tumor growth and angiogenesis, were expressed at significantly higher levels in SMC_1. IHC results validated that the protein expression of FAP, BMP1, WNT5A were upregulated in tumor (Fig. 3h). Hereafter, we defined cells in the three tumor−enriched cell clusters, Endo_1, Fib_1, and SMC_1 as tumor-associated stromal cells (TASCs).

Gene set variation analysis found that genes expressed in TASCs exhibited distinct metabolic signatures and broadly participated in cancer-related pathways (Supplementary Fig. 4f). Notably, angiogenesis pathway, one of the key signatures of tumor progression, was significantly upregulated in TASCs (Fig. 3k). We then examined potential associations between the fraction of TASCs and the survival of patients in TCGA-STAD dataset. The cell type proportions in TCGA-STAD were estimated by MuSiC[27], an algorithm to implement bulk tissue cell type deconvolution with scRNA-seq data. Remarkably, we found that both of Fib_1 and SMC_1 was associated with a worse prognosis (Fig. 3l), and individual genes such as *INHBA* and *PLXDC1* specifically expressed by Fib_1 and/or SMC_1 also held potential prognostic capability (Supplementary Fig. 4g, i). Kumar et al. reported that recombinant INHBA was sufficient to upregulate the expression of FAP and collagen genes in normal fibroblast lines[11]. Besides, our in vitro experiments also showed that CAFs-derived supernatants had the capacity to support tumor growth for several gastric cancer cell lines (Fig. 3m). In summary, TASCs underwent substantial remodeling and displayed potential tumor-promoting features in GC.

### Lipid-associated macrophages were enriched in tumors

Myeloid cells are highly heterogeneous immune cell populations and provide major contributions to shaping the TME[28]. We identified eight distinct clusters of myeloid cells in the GC TME, including two monocyte clusters, two macrophage clusters, and four dendritic cell (DCs) clusters (Fig. 4a, b). We classified the two blood-enriched clusters of cells, Mono_CD14 and Mono_FCGR3A, as classical *CD14+CD16-* and non-classical *CD14-CD16+* monocytes, respectively (Supplementary Fig. 5a, b).

The Mφ_THBS1 showed comparable enrichment in both tumor and paratumor tissues, and expressed *IL1B*, *NLRP3*, *VEGFA* and *EREG*, similar to the pattern of resident tissue macrophages (RTMs) in colon cancer characterized by Zhang et al.[29] Meanwhile, Mφ_APOE cluster, preferentially enriched in tumor tissue, was similar to tumor-associated macrophages (TAMs) in hepatocellular carcinoma (HCC) and the lipid-associated macrophages in adipose tissue by expressing *APOE*, *TREM2*, *C1QA*, and *GPNMB*[30,31]. We further performed multicolor IHC staining to validate the presence of the two distinct macrophages subtypes on tumor sections from GC patients (Supplementary Fig. 5c). Correlation analysis showed that the transcriptional profile of Mφ_THBS1 was close to Mono_CD14, while Mφ_APOE was dissimilar to any other clusters (Supplementary Fig. 5d). We found that Mφ_APOE co-expressed features of both M1 and M2 macrophages[32] (Fig. 4c), indicating that the classical polarization model might not be suitable for evaluating the state of Mφ_APOE in GC.

By dissecting the DEGs between Mφ_APOE and Mφ_THBS1, we found that both lipid-related genes (*APOE*, *TREM2*) and lysosomal genes (*GRN*, *CD63*, *LAMP1*) were highly expressed by Mφ_APOE, and were specifically elevated in the tumor (Fig. 4d, e), indicating lipid-associated and lysosome functions were key identifiers of macrophage in GC (Fig. 4c). We next tried to identify the candidate regulators of Mφ_APOE cells using SCENIC (Supplementary Fig. 5f). Of note, *MITF*, *NR1H3* and *TFEC* were specifically upregulated in Mφ_APOE, and the AUCell scores of the regulons also exhibited similar patterns to the expression of lipid-associated and lysosomal genes (Fig. 4f and Supplementary Fig. 5g). *MITF* was previously found to be involved in lysosomal biogenesis[33]. *NR1H3*, encoding liver X receptor alpha (LXR-alpha), was reported to be involved in the regulation of *APOE* expression in macrophages and adipocytes[34]. To validate these findings, we overexpressed *TFEC* and *NR1H3* in THP-1 monocytes-derived macrophages, which was confirmed by western blot (Supplementary Fig. 5h). Compared with control, we found *TFEC/NR1H3* overexpression significantly upregulated the expression of *APOE* and *APOC1* in macrophage under different stimulations (Fig. 4g). In summary, our analysis suggests that dysregulation in lipid and lysosome-associated functions are hallmarks of TAMs in GC.

In addition to the three traditional DC cell types, cDC1_XCR1, cDC2_CD1C, and pDC_LILRA4, we also found one non-classical DC cell type, DC_LAMP3, characterized by the specific expression of *LAMP3* and *CCR7* (Fig. 4b). LAMP3+ DCs was also detected in hepatocellular carcinoma and other cancer types[30,35], and was capable of migrating from tumor to lymph nodes (Fig. 4c). Based on the RNA velocity analysis, we proposed that LAMP3+ DCs were likely derived from cDC2 (Fig. 4h and Supplementary Fig. 5i). Although cDC2_CD1C cells

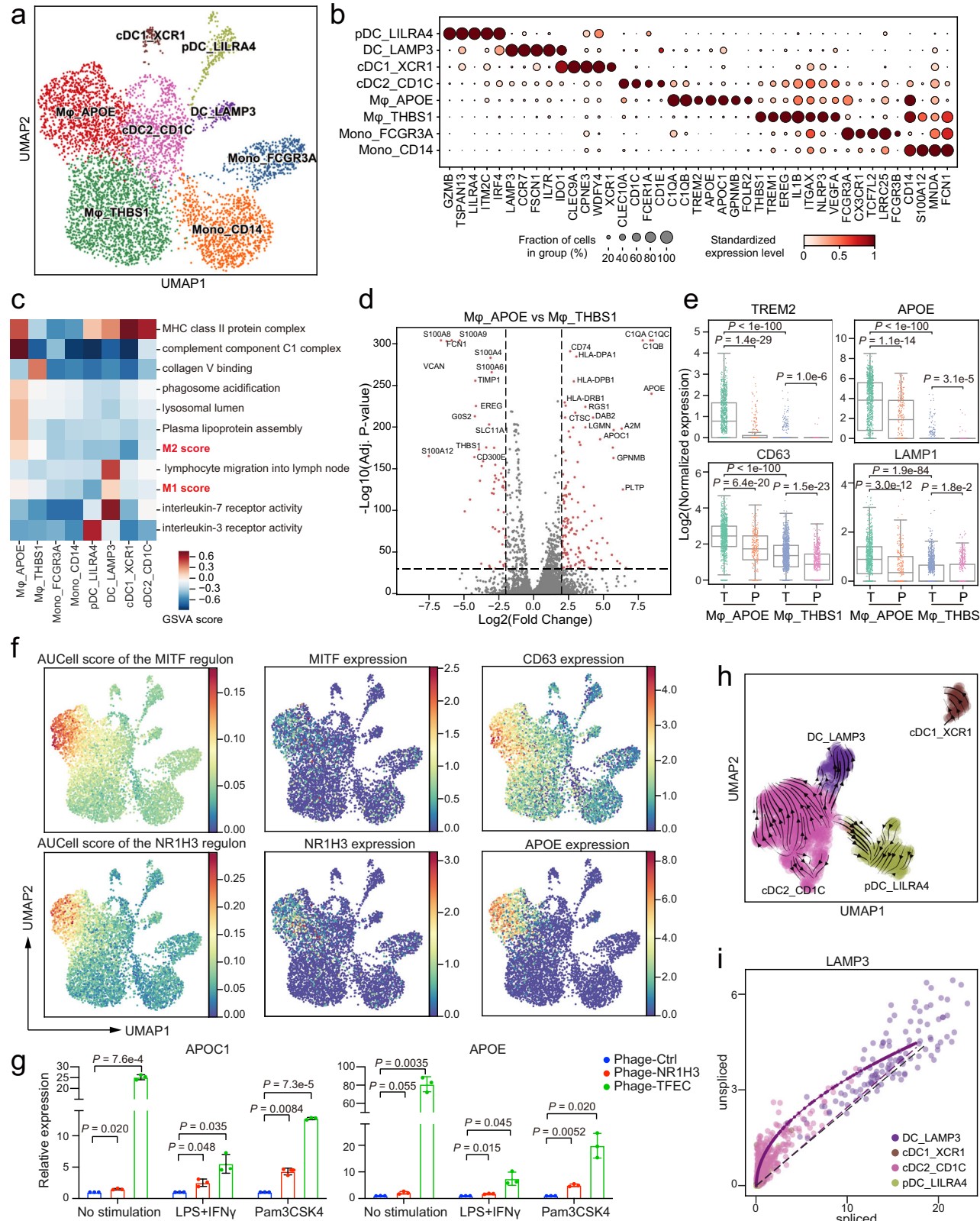

expressed extremely low levels of *LAMP3* and *CCR7*, the unspliced RNAs of *LAMP3* and *CCR7* were relatively high (Fig. 4I and Supplementary Fig. 5j).

## Tc17 cells are present in the TME of most gastric tumors
To dissect the diversity of T cells in GC, we further extracted and reclustered T cells that had both scRNA-seq data and paired TCR

information into ten CD8⁺ clusters, six conventional CD4⁺ clusters (CD4⁺ Tconv), three CD4⁺ Treg clusters, and one cycling cluster which represented T cells currently progressing through the cell cycle (Fig. 5a–c and Supplementary Fig. 6).

Both CD8_C1_LEF1 and CD4_C1_CCR7 clusters were found to express naïve marker genes such as *LEF1* and *CCR7*, indicating their naïve state. In comparison, CD8_C2_CX3CR1 cells expressed genes

**Fig. 4 | Dissection of myeloid cells showing the expansion of lipid-associated macrophages in tumor tissues. a** UMAP of myeloid cells. Clusters are labeled with inferred cell types. **b** Dot plots showing marker genes across myeloid cell subsets. Dot size indicates the proportion of expressing cells, colored by standardized expression levels. **c** Differences in pathway activities scored by GSVA among the different myeloid cell clusters. The scores of pathways are z-score-normalized. **d** Volcano plot showing differentially expressed genes between Mφ_APOE and Mφ_THBS1. Dotted lines indicate *p*-value < 1e−20 and |log₂(FC)| > 2 (two-sided Wilcoxon rank-sum test). **e** Boxplot showing lysosomal and lipid-related gene expression in macrophages. Each dot represents a single cell. T and P represent tumor and paratumor tissues, respectively. (two-sided

Wilcoxon rank-sum test, $n = 1236$ for Mφ_THBS1_T, $n = 926$ for Mφ_APOE_T, $n = 544$ for Mφ_THBS1_P, $n = 244$ for Mφ_APOE_P). **f** UMAP of myeloid cells, colored by the AUCell scores of the TF regulon activity of NR1H3 and MITF, or by the normalized expression of genes. **g** NR1H3 or TFEC overexpressed THP-1-derived macrophages were stimulated with lipopolysaccharide (LPS) + interferon γ (IFNγ) or Pam3CSK4. The expressions of *APOE* and *APOC1* were then measured by qPCR. Each column represents the mean ± SD of three independent experiments (two-sided *t*-test). **h** UMAP showing the inferred development dynamics of DC subsets by RNA velocity. **i** Velocity analysis of the spliced and unspliced mRNAs of *LAMP3* in DCs. Each dot represents one cell. Source data are provided as a Source Data file.

associated with effector function (*FCGR3A*, *FGFBP2*). CD8_C3_GZMH was characterized by high expression of cytotoxic genes (*GZMK*, *GZMH*), and MHC-II genes (*HLA-DRA*, *HLA-DRB5*), which are recognized as markers of T cell activation. The CD8_C4_GZMK cluster was characterized by high expression of the *GZMK*, *CD44*, and *CXCR4* genes, commonly associated with effector memory cells. We also found a strong correlation between the transcriptional profiles of CD8_C5_TOB1 and CD8_C6_GNLY (Supplementary Fig. 7a), both of which were comprised of tissue-resident memory CD8+ T cells, but can be clearly marked by the expression of *TOB1/CXCR6/ANXA1*, and *GNLY/XCL1/XCL2*, respectively (Supplementary Fig. 6c). The three CD4+ Tconv clusters, CD4_C2_LTB, CD4_C3_SLC2A3, and CD4_C4_CD69, were respectively associated with blood central memory (*S1PR1*, *ICAM2*), tissue central memory (*TCF7*, *GPR183*, *CXCR4*), and tissue-resident memory (*MYADM*, *RGS1*, *CD69*) (Supplementary Fig. 6d). CD8_C9_HAVCR2 and CD4_C5_CXCL13 differentially expressed immune checkpoint genes such as *CTLA4*, *PDCD1*, and *TIGIT*, known markers of T cell exhaustion. CD8_C7_CD160 had a high expression of natural killer cell marker genes (*KLRC1*) and *CD160*, suggesting that this cluster was composed of intraepithelial lymphocytes (IELs). CD8_C10_SLC4A10 was characterized as mucosal-associated invariant T cells (MAITs) based on the specific expression of *SLC4A10*, *RORA*, and *TRAV1-2*[36]. Three Treg subtypes, including Treg_C1_SELL, Treg_C2_LAG3, Treg_C3_CTLA4, exhibited characteristic co-expression of *CD4*, *FOXP3*, and *IL2RA*, but were further distinguished by cluster-specific expression of naïve markers (*LEF1*, *SELL*), follicular regulatory T cells markers (*IL10*, *CXCR5*), and suppressive Treg markers (*CTLA4*, *CCR8*).

Surprisingly, in addition to CD4_C6_IL17A (Th17), we found that the CD8_C8_IL17A cluster also highly expressed several known classical markers of Th17 cells, such as *IL17A*, *RORC* (RORγt), and *IL23R*. We observed that CD8+IL17+ T cells constituted more than 1% of the total tumor infiltrated T cell population in 8 out of the 10 patients (Supplementary Fig. 6g), and the presence of CD8+IL17+ T cells in the TME of GC were subsequently confirmed by multicolor IHC staining (Fig. 5d and Supplementary Fig. 6e). The two IL17A+ clusters, despite belonging to the two distinct classical lineages of T cells (i.e., CD4+ and CD8+ T cells), had a highly correlated patterns of gene expression (Supplementary Fig. 7a). Enrichment analysis showed that the upregulated genes in CD8_C8_IL17A were significantly enriched in Th17 cell differentiation and inflammatory bowel disease (IBD) pathways, highly similar to that of Th17 cells (Fig. 5e and Supplementary Fig. 6f). These results suggested CD8_C8_IL17A and Th17 cells might have overlapping functions in the TME of GC. These characteristics of the CD8_C8_IL17A cells were akin to Tc17 cells[37], leading us to designate this cluster Tc17.

Notably, several clusters of T cells displayed distinct patterns of tissue distribution (Fig. 5b and Supplementary Fig. 7b). For instance, naïve T cells were predominantly enriched in the blood; resident memory (CD8_C5), IELs, and follicular regulatory T cells were specifically enriched in paratumor tissue, while exhausted CD8+ T cells, IL-17+ T (Tc17, Th17) cells, and suppressive Treg were largely prevalent in tumors.

## Tc17 and Th17 cells may promote tumor growth via IL17/22/26 signaling in GC

In the past decade, a group of specialized CD8+ T cells with distinct cytokine production, Tc17 (CD8+IL17+ T) cells, were detected in multiple types of cancers in the gastrointestinal system and were found to be associated with poor survival in patients with such cancers[37–40]. Both the proportion of Tc17 cells and the level of cytokines produced by Tc17 cells are reported to be negatively associated with the survival of GC patients[38,41]. An attempt to understand the function of Tc17 cells in gastric cancers using ex vivo experiments showed that Tc17 cells isolated from the gastric tumor can stimulate tumor cells to produce CXCL12, which consequently recruit myeloid-derived suppressor cells (MDSCs) to suppress cytotoxic CD8+ T cells[38].

Among the 26 genes that were significantly upregulated (FC > 2, *p*-value < 0.01) in both Th17 and Tc17 cells (Fig. 5f and Supplementary Data 4), *IL17A*, *IL17F*, *IL22*, and *IL26* were reported to facilitate tumor progression through multiple mechanisms[42–44]. Interestingly, we found their receptors, *IL17RA/IL17RC* for *IL17A/IL17F*, *IL10RB/IL22RA1* for *IL22*, and *IL20RA/IL10RB* for *IL26*, were upregulated in tumor cell than in normal epithelial cells (Fig. 5g). Consistent with our findings in the scRNA-seq data, these receptors were also expressed higher in tumor tissues than in normal tissues of the TCGA-STAD patients (Fig. 5h). Thus, we hypothesized that IL17+ T cells in GC could promote tumor growth via *IL17*, *IL22*, and *IL26* signaling. Besides, TASCs such as, Fib_1 and SMC_1, also expressed higher level of IL17RA/IL17RC and might be regulated by Tc17/Th17.

Previous studies showed that binding of CEACAM5, also known as carcinoembryonic antigen (CEA), on intestinal epithelial cells to CD8a on T cells can induce a group of regulatory CD8+ T cells (CD8+CD28−CD101+CD103+), which are not cytotoxic cells and can inhibit proliferation of CD4+ T cells[45,46]. We found that the expression of surface markers in Tc17 cells matched that of the aforementioned regulatory CD8+ T cells almost perfectly (Supplementary Fig. 7c). Moreover, Tc17 cells expressed the highest level of *ITGAE* (CD103), a member of the integrin family, among all subtypes of T cells detected in the single-cell analysis (Supplementary Fig. 7d), suggesting that Tc17 cells may involve in physical cell-cell contact with epithelial cells via ITGAE-CEACAM5 interaction.

## Dissecting state transition of T cell subtypes by TCR analysis

Next, we sought to understand the cell state transitions among various subtypes of T cells using TCR clonal information and pseudotime analyses. Among the 36,239 clonotypes found in T cells, 30,980 were only detected once (unique TCR), while 5259 were detected in two or more T cells (non-unique TCR). The individual clone population sizes ranged from 1 to 569 (Fig. 6a). In general, CD8+ clusters had a higher degree of clonal expansion[47] than CD4+ clusters except for naïve CD8+ T cells (CD8_C1) (Fig. 6b and Supplementary Fig. 8a, b). We observed that CD8_C2 (Effector), CD8_C3 (Cytotoxic), CD8_C4 (Effector memory), and CD8_C10 (MAIT) clusters had both higher proportions of clonal cells and proportions of clonal cells with TCR shared between blood and solid tissue (Fig. 6c). Among the high clonality clusters, CD8_C2 was mostly derived from blood and its marker genes were

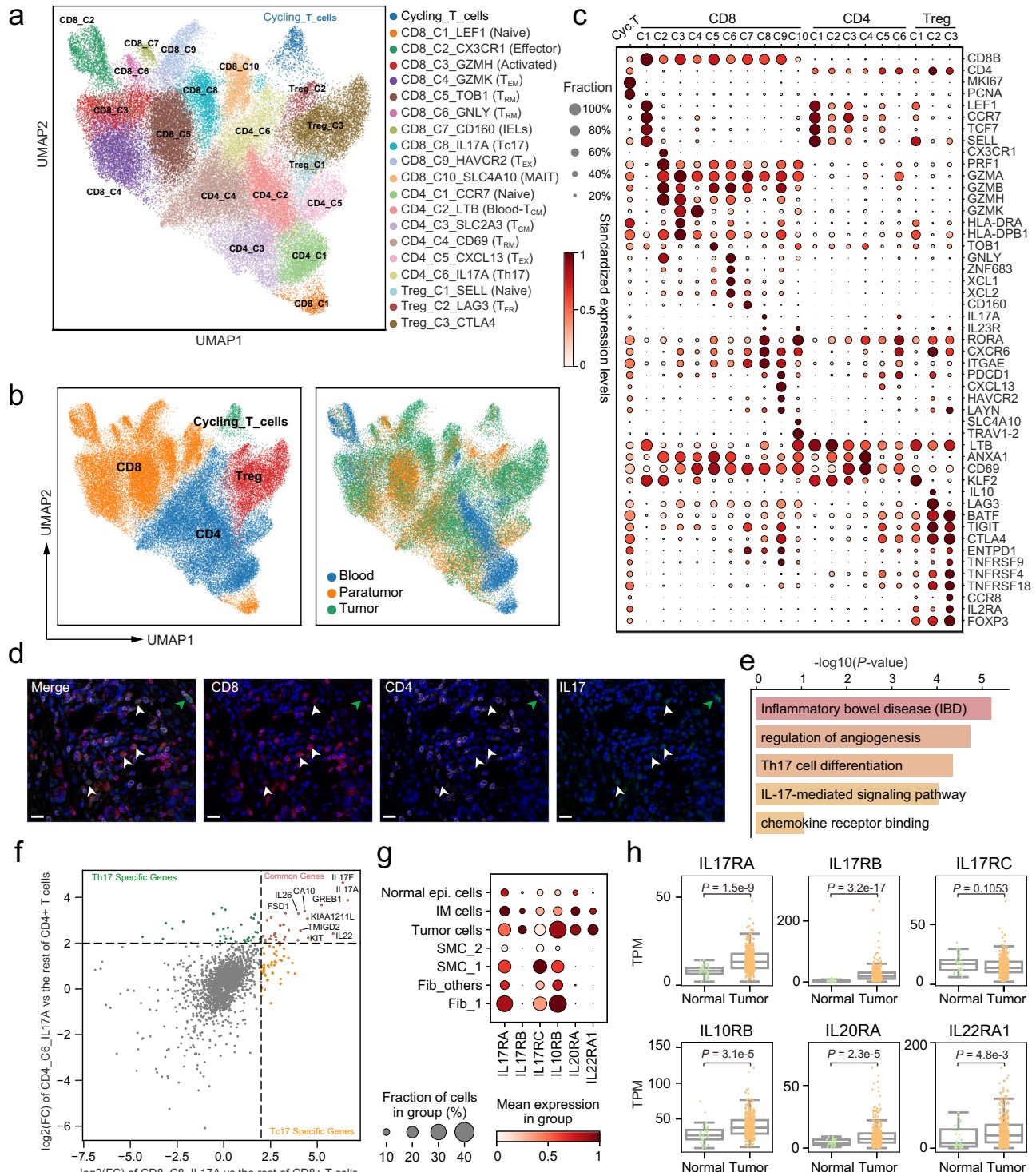

**Fig. 5 | Dissection and clustering of T cells implicating multiple distinct functional states and gene modules in GC ecosystem. a** UMAP of T cells that had both scRNA-seq data and paired TCR information. Clusters are labeled with inferred cell types. **b** Dot plots showing marker genes across T cell subsets. Dot size indicates the proportion of expressing cells, colored by standardized expression levels. **c** UMAP of T cells colored by cell type (left) and cellular tissue origin (right). **d** Multicolor IHC staining with anti-CD4, anti-CD8, and anti-IL17A antibodies, exemplified by patient GC988401 ($n = 6$). The white and green arrows indicate CD8$^+$IL17$^+$ cells and CD4$^+$IL17$^+$ cells, respectively. The scale bar represents 20 μm. **e** Bar plot of the KEGG terms or pathways enriched for highly expressed genes in Tc17, the *p*-values were calculated by the hypergeometric distribution. **f** Scatterplot showing the log₂ fold change of differentially expressed genes. CD8_C9_HAVCR2 versus other CD8$^+$ T cells (X-axis); Treg_C3_CTLA4 versus other CD4$^+$ T cells (Y-axis). Each dot represents a gene, with color annotation inside. **g** Dot plots showing the expression of genes encoding the receptors for *IL17A*, *IL17F* (*IL17RA/IL17RC*), *IL22* (*IL10RB/IL22RA1*), and *IL26* (*IL20RA/IL10RB*) in scRNA-seq dataset. **h** Box plots showing the expression of genes encoding the receptors for *IL17A*, *IL17F* (*IL17RA/IL17RC*), *IL22* (*IL10RB/IL22RA1*), and *IL26* (*IL20RA/IL10RB*) in tumor ($n = 375$) and normal tissue ($n = 32$) of TCGA-STAD dataset, grouped by tissue origin (two-sided Wilcoxon rank-sum test). Source data are provided as a Source Data file.

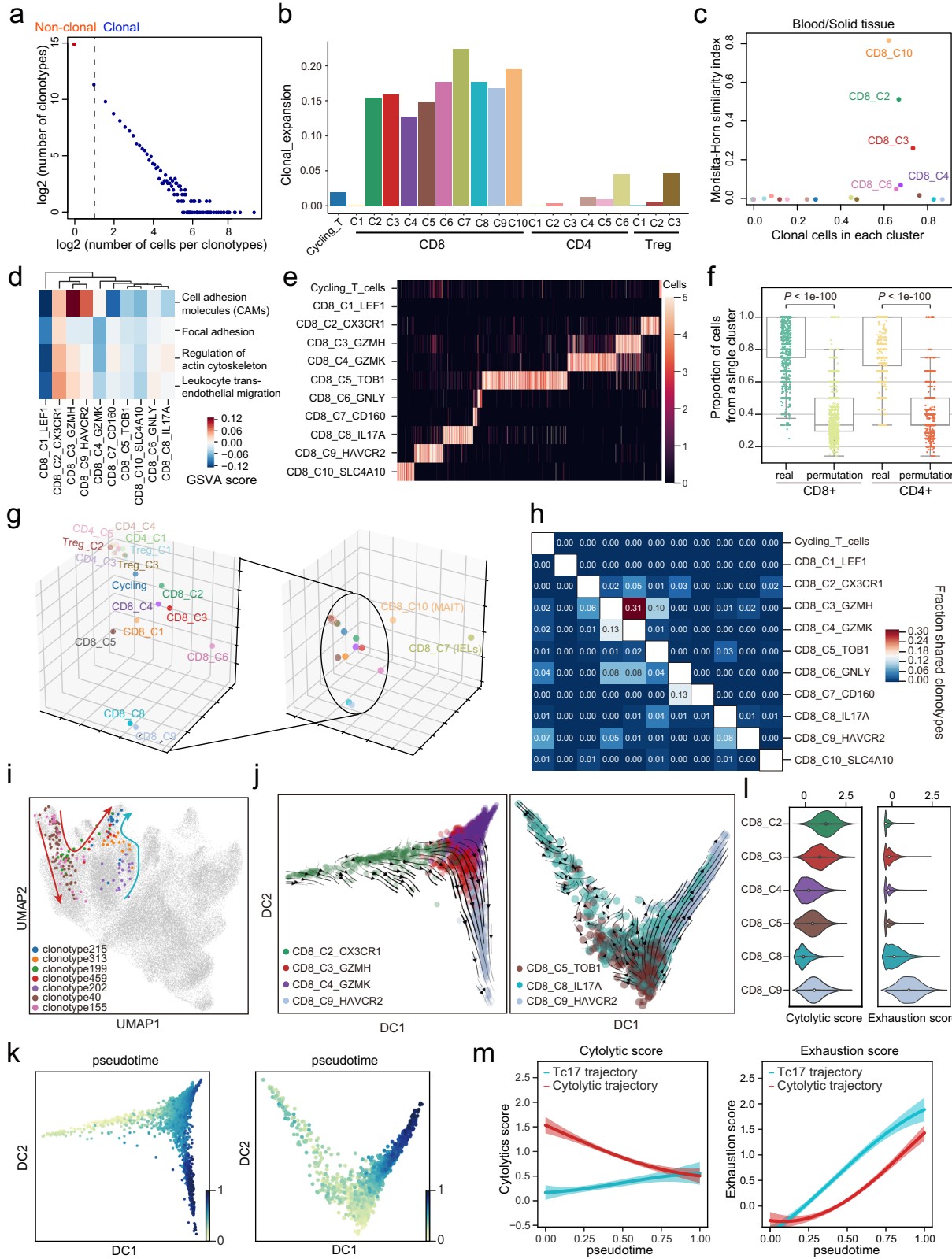

enriched in cell migration-related pathways (Fig. 6d), therefore we speculated that CD8_C2 had the potential to infiltrate into the solid tissue from the blood.

Consistent with the result of a recent study[48], we found that T cells of the same clonotype were more likely to aggregate in the same cluster or in closely related clusters rather than uniformly distributed in all clusters (Fig. 6e, f and Supplementary Fig. 8c−e). With the single-

cell TCR sequencing, we embedded the T-cell clusters into a 3D dimension space according the TCR-gene usage within each cluster (Fig. 6g and Supplementary Data 5). The embedding space suggested CD4⁺ T cells and CD8⁺ T cells had different biases on TCR-gene usage and CD8⁺ T cells showed greater intra-lineage heterogeneity than CD4⁺ T cells. Among CD8⁺ T-cell clusters, CD8_C10 (MAIT) and CD8_C7 (IELs) were extremely far from the other clusters in the space, indicating

**Fig. 6 | Phenotype transition of CD8+ T cells based on both TCR sharing and trajectory analysis. a** The association between the number of T cell clones and the number of cells per clonotype. The dashed line separates non-clonal and clonal cells, with the latter identified by repeated usage of TCRs. **b** Bar plot showing the score of clonal expansion in each T cell subset. **c** Comparison between the proportions of clonal cells (x-axis), and the clone sharing between blood and solid tissues estimated by Morisita–Horn similarity index (y-axis) in each cluster. **d** Differences in migration-related pathway activities scored by GSVA among different CD8+ T cell subsets. **e** The distribution of clonal clonotypes in CD8+ T-cell subsets and cycling T cells. Lighter color indicates higher cell number; Cell numbers were capped at 5. **f** Proportions of cells from a single cluster for each clonotype. Permutation was performed among cluster labels and clonotype labels (two-sided Wilcoxon rank-sum test, *n* = 1558 for both real and permuted CD8+ T cell clonotypes, *n* = 1046 for both real and permuted CD4+ T cell clonotypes). **g** Three-

dimensional plots showing the PCA embedding of T-cell subsets according to the bias in VDJ-gene usage of each cluster. **h** Heat map showing the fraction of clonotypes belonging to a primary phenotype cluster (rows) that are shared with other secondary phenotype clusters (columns). **i** UMAP of T cells colored by selected TCR clonotypes. Red and cyan arrows indicate the state transition of blood-derived and tissue-derived CD8+ T cells, respectively. **j** Diffusion maps showing the RNA velocity of cells from the cytolytic trajectory (left) and the Tc17 trajectory (right). **k** Diffusion maps showing the pseudotime of the cytolytic trajectory (left) and the Tc17 trajectory (right), which were calculated according to RNA velocity. **l** Violin plot showing the cytolytic score (left) and the exhaustion score (right). **m** Gaussian process regression curves with a 95% confidence interval showing the dynamics of the cytolytic score (left) and the exhaustion score (right) along the pseudotime of the cytolytic trajectory (excluding CD8_C4_GZMK) and the Tc17 trajectory, respectively. Source data are provided as a Source Data file.

unique VDJ-gene usages in these two clusters. CD8_C10 (MAIT) had a highly conserved usage of TRAV1-2 and a relatively high usage of TRAJ33 in accordance with previous studies[36], while CD8_C7 (IELs) show no significant bias on VDJ-gene usage.

We then systematically evaluated relationships in their lineages by calculating a clonotype-sharing matrix based on the proportion of TCR clonotypes shared by pairwise cell clusters (Fig. 6h and Supplementary Fig. 8e, f). As expected, we noted that the matrix of CD4+ T cells was sparse compared to that of CD8+ T cells, since the clonal expansion in CD4+ T cells was much lower (Fig. 6b). Interestingly, we identified CD8_C8 (Tc17), in addition to CD8_C3 (Cytotoxic), as a potential source of CD8_C9 (Exhausted). Given the differences in the clonotype-sharing matrix (Fig. 6h) and the shared proportions between blood and solid tissue (Fig. 6c), we thus hypothesized two possible trajectories for T cell state transition (Fig. 6i) that depicted the fates of blood-derived CD8+ T cells and tissue-resident CD8+ T cells, respectively. In this model, we postulated that CD8_C3 (Cytotoxic) cells were derived from CD8_C2 (Effector; Blood origin) and could become CD8_C4 (Effector memory) or CD8_C9 (Exhausted); alternatively, CD8_C8 (Tc17) derived from CD8_C5 (Resident; Tissue origin) cells could transition into CD8_C9 (Exhausted) cells.

### Resident CD8+ T cells can reach exhaustion through the Tc17 trajectory

To further decipher the differentiation trajectories across these clusters, we first extracted cells that shared the same clonotypes among the clusters in each potential trajectory, and then we used RNA velocity analysis to interrogated their directionality embedded on a diffusion map[49] (Fig. 6j, k and Supplementary Fig. 9a, b). We identified a strong directional stream from blood-derived CD8+ T cells to the exhausted population via CD8_C3 (Cytotoxic) cells (Fig. 6j, left). Along this trajectory to exhaustion, the cytolytic scores of T cells gradually decreased and the exhaustion scores gradually increased (Fig. 6l, m). Consistent with the result of the clonotype-sharing analysis, RNA velocity showed tissue-resident CD8+ T cells also exhibited a directional stream toward the exhausted population via Tc17 cells (Fig. 6j, right), suggesting tissue-resident CD8+ T cells could differentiate into Tc17 cells in the TME, and subsequently give rise to exhausted phenotype.

Henceforth, we named the two trajectories of T-cell exhaustion "the cytolytic-exhaustion trajectory" and "the Tc17-exhaustion trajectory". Despite reaching the exhausted state in the end, we found that the exhaustion scores were significantly elevated along the Tc17-exhaustion trajectory compared to that of the cytolytic-exhaustion trajectory (Fig. 6m). Moreover, Tc17 has the lowest cytolytic score in non-naïve CD8+ T cells despite undergone extensive clonal expansion, while CD8_C2 (Effector) T cells ranked first in the list (Fig. 6b and Supplementary Fig. 9c). Overall, the cytolytic score was decreasing along the cytolytic-exhaustion trajectory and increasing along the Tc17-exhaustion trajectory (Fig. 6m). In summary, our observations

suggest that tumor-infiltrating CD8+ T cells can reach the exhausted state through both cytotoxic T cells and Tc17 cells in the TME of GC.

### Distinct transcription programs are associated with the two exhaustion trajectories

Tc17 cells differ from cytotoxic T cells (CD8_C3) not only in their transcriptional profile (Fig. 5a and Supplementary Fig. 7a) but also in the VDJ gene usages of TCR (Fig. 6g), suggesting that these two subtypes of T cells may recognize two different sets of antigens and are functionally distinct. We speculated that Tc17 cells and cytotoxic T cells could give rise to two separate types of exhausted T cells with distinct transcriptional programs. Supporting this notion, differential gene analysis showed that Tc17-derived exhausted T cells highly expressed keratin *KRT86*, while cytolytic-cell-derived exhausted T cells highly expressed *GZMK* (Fig. 7a, b and Supplementary Fig. 9d).

To investigate the underlying mechanism driving the distinct transcriptional programs of the two trajectories (Fig. 7c), we focused our analyses on dynamics of transcription factors (TFs) that are differentially expressed and activated in the two paths. Along the cytolytic-exhaustion trajectory, we found many classical TFs were expressed at the exhausted state, such as *PRDM1* and *TOX2* that were known to promote T cell exhaustion. While the expressions of some TFs like *EOMES* were high at the beginning, extended over the cytotoxic state, and disappeared at the end of the exhausted state. In the exhaustion period of the trajectory, we also observed expression of TFs less described in the context of T cell exhaustion, including *NR3C1*, *CEBPD*, and *ATF3*. Along the Tc17-exhaustion trajectory, we found *RORA* and *RORC*, classical master regulators in Th17 cells, were expressed in Tc17 cells, suggesting both TFs participated in maintaining the cellular state of Tc17 cells. TFs like BHLHE40 and CREM expressed at the end of the Tc17-exhaustion trajectory may promote the exhausted state of Tc17 cells.

We next used SCENIC to identify TFs that differentially activated along the two trajectories (Fig. 7d). Through SCENIC, combined with the pseudotime and gene expression analysis, we found that both the expression of EOMES and its downstream targets were high in the cytolytic-exhaustion trajectory compared to that of the Tc17-exhaustion trajectory (Fig. 7c–f). Similarly, RUNX2 was identified as a potential key regulator in the Tc17-exhaustion trajectory due to the high expression of these factors and their downstream targets (Fig. 7c–f). To summarize, our analyses highlighted the distinct regulatory program along the two exhaustion trajectories and identified potential key modulators of the two processes.

### Tumor-associated stromal and myeloid cells are key mediators of the complex cellular interaction

In order to dissect the complex network of communication among the various cell types that participate in GC, we next identified putative cell-cell interactions in both tumor and normal tissue by CellPhoneDB[50]. Obviously, interactions involving TASCs and

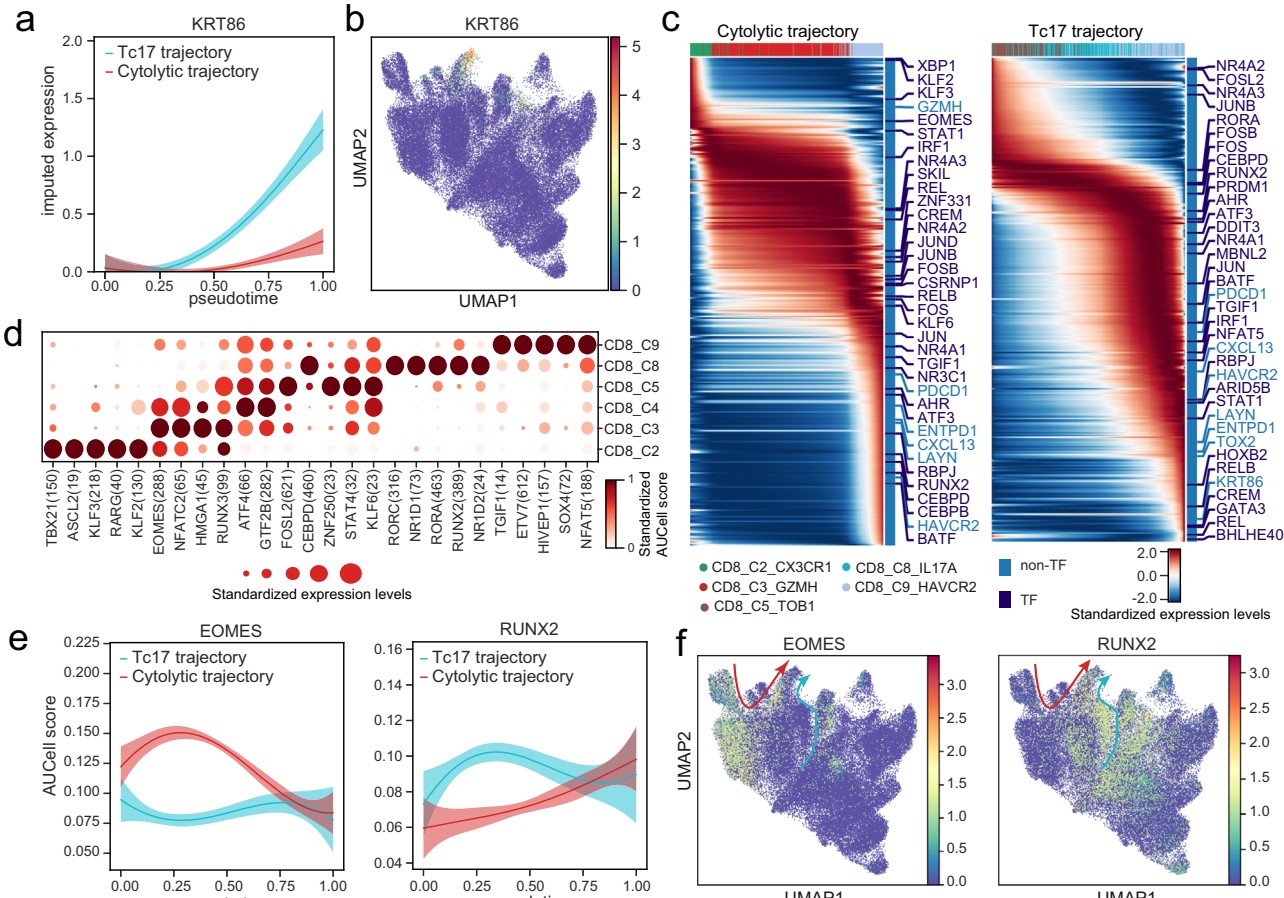

**Fig. 7 | Dynamics of TF activities along the T-cell exhaustion trajectories and the potential promoting effect of IL17⁺ T cells on gastric tumors. a** Gaussian process regression curves with a 95% confidence interval showing the dynamic expression of *KRT86* along the pseudotime of the cytolytic trajectory (excluding CD8_C4_GZMK) and the Tc17 trajectory. **b** UMAP showing the expression of *KRT86* in T cells. **c** Heatmaps showing the expression of highly variable genes along the pseudotime of the cytolytic trajectory (excluding CD8_C4_GZMK) and the Tc17 trajectory. The color bar on the top represents cell clusters as in Fig. 6g; The color bar on the right annotates all the highly variable TFs and some specific marker genes. **d** Dot plot showing the AUCell scores of TF regulon activity calculated by SCENIC for CD8⁺ T-cell subsets. The size of each dot represents the standardized expression level of TFs. **e** Gaussian process regression curves with a 95% confidence interval showing the dynamic expression of *EMOES* (left) and *RUNX2* (right) along the pseudotime of the cytolytic trajectory (excluding CD8_C4_GZMK) and the Tc17 trajectory. **f** UMAP showing the expression of *EMOES* (left) and *RUNX2* (right) in T cells. Red and cyan arrows indicate the state transition of blood-derived and tissue-derived CD8⁺ T cells, respectively. Source data are provided as a Source Data file.

macrophage cells dominated the TME networks (Supplementary Fig. 10a, b and Supplementary Data 6).

Focusing on TASCs, we found tumor cells had more interactions with stromal cells than did normal or tumor-like epithelial cells (Supplementary Fig. 10c). Fib_1 expressed a wealth of growth factors for tumor cells, such as *HGF, FGF7*, and *BDNF* (Fig. 8a), in agreement with other studies wherein the *HGF-MET* and *FGF7-FGFR4* axes were shown to be promising therapeutic targets for GC and other tumors[51,52]. Notably, we found that *HGF* was negatively associated with survival in the TCGA-STAD dataset (Supplementary Fig. 10d). Furthermore, VEGFA-dependent angiogenesis and ephrin-Eph bidirectional signaling pathways were also found in Endo_1 and tumor cells. Remarkably, TASC subtypes exhibited intimate signaling networks (Fig. 8b). For example, *TEK* expressed by Endo_1, is the receptor for *ANGPT2* expressed by SMC_1, suggesting that SMC_1 is involved in regulating endothelial cell survival and migration[53]. In addition, TASCs expressed high levels of Notch ligands *JAG1* and *DLL1* that interact with Notch receptors *NOTCH1, NOTCH3*, and *NOTCH4* on Endo_1 and/or SMC_1, which is consistent with the results of the GO function enrichment analysis (Supplementary Fig. 10e). Endo_1 demonstrated strong activation of the TNF, VEGF, PDGF, PGF, and Notch signaling pathways (Fig. 8a, b), which are widely involved in the biological process of angiogenesis[54]. Notably, TASCs were the key supplier for cytokines in these pathways (Fig. 8b). This analysis further explained the cause of upregulation of angiogenesis in TASCs (Fig. 3k). By performing the correlation analysis among each cell type, we found that the proportions of TASC subtypes were highly positive correlated in both our dataset and TCGA-STAD cohort (Fig. 8c and Supplementary Fig. 11b–d).

Further, we analyzed the molecular interactions between Mφ_APOE and TASCs, which had a dominant interaction in GC (Fig. 8a and Supplementary Fig. 10a, b). Mφ_APOE was predicted to interact with TASCs via *OSM, IL6, IL1B*, and *TNF*, which are key activators of stromal cells[55]. Secretion of *CSF1* and *IL34* by SMC_1 and/or Fib_1, and their interaction with *CSF1R* on Mφ_APOE were previously reported to contribute to survival, proliferation, and differentiation of macrophages and monocytes[56]. Consistently, the expression of *IL34* was negatively associated with survival in the TCGA-STAD dataset (Supplementary Fig. 10d). Endo_1 and Fib_1 both expressed high levels of *PROS1*, which interacts with *AXL* expressed by Mφ_APOE to impair the antitumor immune response by macrophage[57]. Interestingly, the proportion of Mφ_APOE also demonstrated positive correlations with TASCs in both our dataset and TCGA-STAD (Fig. 8c, and Supplementary Fig. 11b). Moreover, our in vitro co-culture experiment evinced

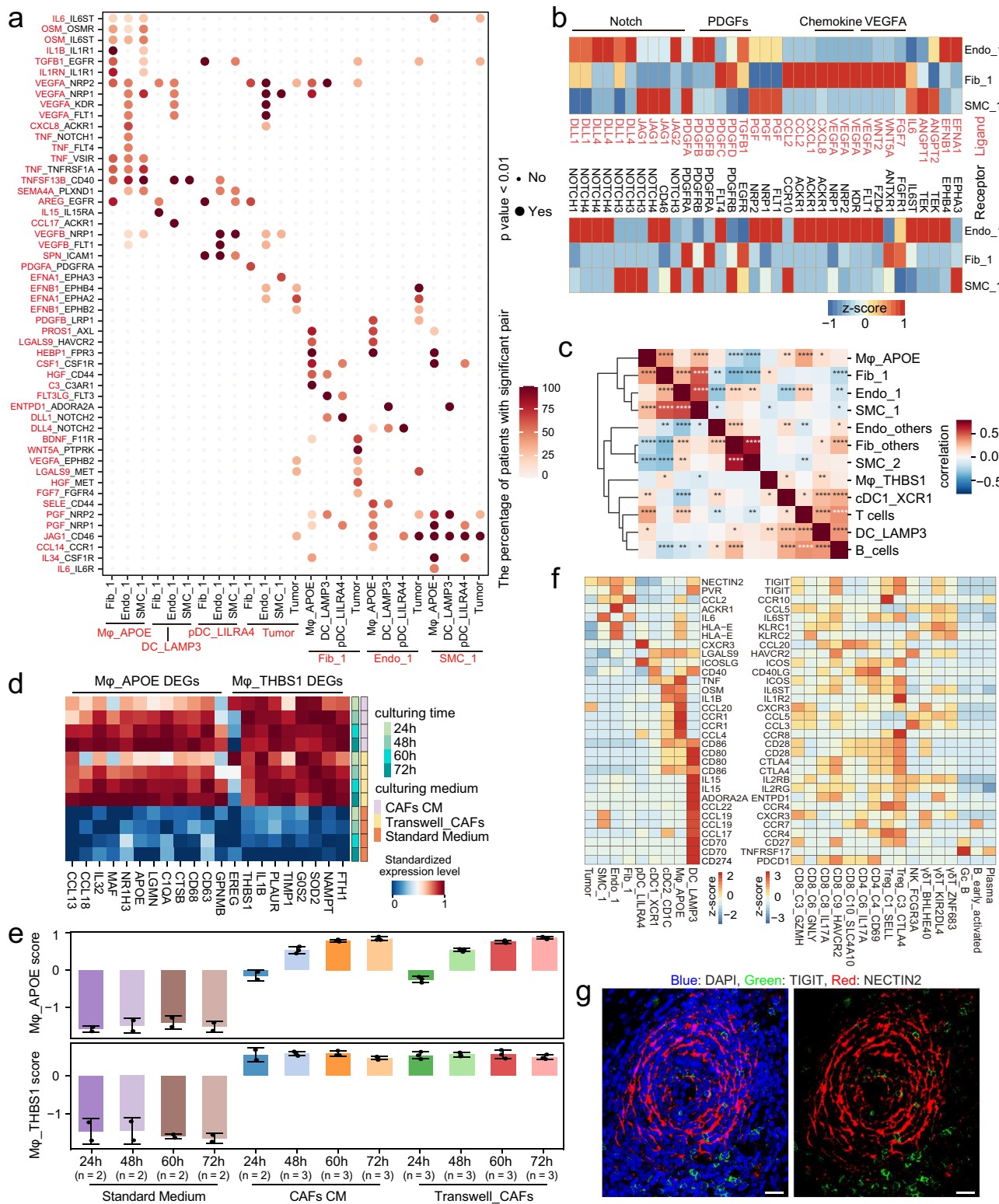

that certain cytokine secreted by CAFs could induce THP-1 monocyte-derived macrophages into the states of macrophages in the GC TME, upregulating the marker genes of both Mφ_APOE and Mφ_THBS1 (Fig. 8d). The Mφ_APOE score showed a continuous increasement, while the Mφ_THBS1 score displayed slight variation during the induction (Fig. 8e). We also found cytokines interacting with TASCs, such as *IL6* and *OSM*, were highly expressed by induced macrophages (Supplementary Fig. 11e). Together, our results suggested that Mφ_APOE and TASCs, were locked into a mutually reinforcing

feedback loop via several L-R pairs to maintain the pro-tumoral microenvironment.

Next, we investigated significant L-R interactions associated with lymphocytes in TME (Fig. 8f and Supplementary Fig. 11a). *CD80* and *CD86* on myeloid cells were found to provide co-stimulatory signals for CD8+ and CD4+ Tconv cells via *CD28*, while *CTLA4* on Tc17, Th17, Treg, and CD8+ exhausted T cells can act as an antagonist of this interaction by competing for ligand binding with higher affinity[58]. In addition, we identified the *CD40LG-CD40* and *CCL20-CCR6* interaction pairs

**Fig. 8 | The cell-cell interaction networks showed important pathways for tumor progression. a** Dot plot of selected ligand-receptor interactions in tumors. Cell subsets are shown on the x-axis; ligand (red) and receptor (black) pairs are shown on the y-axis. The color of the circle denotes the proportion of patients with a significant interaction (*p*-value < 0.01) in the total patients with these interacting cell subsets. The *p*-values were generated by CellphoneDB which uses a one-sided permutation test to compute significant interactions. **b** Heatmap of scaled expression of selected ligand-receptor pairs for Endo_1, Fib_1, and SMC_1. **c** Heatmap showing the Spearman's rank correlation coefficients between the inferred proportions of different cell types in TCGA-STAD dataset. *$P < 0.05$, **$P < 0.01$, ***$P < 0.001$, and ****$P < 0.0001$ ($P$-value were calculated by two-sided *t*-test and the exact values can be found in the Source Data). **d** Heatmap showing the mean expression of the marker genes of both Mφ_APOE and Mφ_THBS1 in THP-1 monocyte-derived macrophages that were co-cultured with gastric CAFs in a transwell system, cultured by gastric CAFs conditioned mediums (CM), or cultured by standard medium for 24 h, 48 h, 60 h, 72 h. **e** Bar plot showing the Mφ_APOE and Mφ_THBS1 scores. Each column represents the mean ± SD of three duplicates. **f** Heatmap of scaled expression of selected ligand-receptor pairs between lymphocytes (right) and other subsets (left) in tumors. **g** Multicolor IHC staining with anti-TIGIT and anti-NECTIN2 antibodies, exemplified by patient GC769812 (*n* = 6). The scale bar represents 20 µm. Source data are provided as a Source Data file.

between CD4_C4/C6 and DC_LAMP3/cDC1_XCR1, suggesting CD4⁺ T cells were involved in the activation and recruitment of DCs.

Interestingly, DC_LAMP3 cells were predicted to deliver both attracting (*CCL19-CCR7/CXCR3* and *CCL17/CCL22-CCR4*) and activating (*CD70-CD27/TNFRSF17* and *IL15-IL2RB/IL2RG*) signals to lymphocytes, but on the other hand, DC_LAMP3 inhibited anti-tumor T cell activity through a high expression of *CD274* (PD-L1). TASCs, DC_LAMP3, and tumor cells, all highly expressed *NECTIN2* and *PVR* (CD155), whose interaction with *TIGIT* blocks T-cell activation and proliferation[59]. Multicolor IHC staining confirmed the interaction of NECTIN2-TIGIT on tumor sections from GC patients (Fig. 8f and Supplementary Fig. 11f). This analysis suggested that interfering the NECTIN2/PVR-TIGIT axis may serve as a promising therapeutic strategy for the treatment of GC. We found that many types of lymphocytes expressed *ENTPD1* (CD39), which converts ATP into adenosine together with *CD73* to prevent immune activation. Adenosine may subsequently bind with DC_LAMP3 via *ADORA2A* to induce *IDO1* expression (Fig. 4b), which has been found to participate in suppressing effector T and NK cells, as well as differentiation and activation of Tregs and MDSCs[60]. In summary, it is evident that myeloid populations can regulate the state of lymphocytes by complex L-R interactions, and that stromal cells can also inhibit lymphocyte activity via contact-dependent mechanisms.

## Discussion

We present a comprehensive single-cell transcriptome atlas of GC, which described a detailed and complex taxonomy of immune, stromal, and epithelial subsets, and we further illuminated their molecular signature and intercellular communication. The most obvious phenomenon was the extensive remodeling of cellular composition in the TME. Suppressive Tregs, TASCs, TAMs, Tc17, and CD8⁺ exhausted T cells were enriched in the tumor, whereas mast cells, endocrine, and follicular regulatory T cells were enriched in paratumor. Notably, a high proportion of TASCs was associated with a worse prognosis. By ligand and receptor analysis, we observed TASCs, Mφ_APOE and LAMP3⁺ DCs, as the key mediators in complex intercellular networks of interaction, orchestrated the immunosuppressive microenvironment and promoted tumor progression. Blocking these interactions, such as the *IL34-CSF1R* and *TIGIT-NECTIN2* axes, may potentially activate the TME or prime the TME to increase the efficacy of existing immunotherapies in GC.

We observed that Tc17 cells may promote tumor growth via cytokines, *IL17*, *IL22*, and *IL26*. However, both *IL22* and *IL26* were expressed at very low level in bulk sequenced tumor samples of the TCGA-STAD dataset (Supplementary Fig. 7e). Such discrepancy can be explained if Tc17 cells physically interact with tumor cells, therefore, the influence of the cytokines produced by Tc17 cells was profound in the vicinity of epithelial cells despite the overall low expression in the tumor tissue, highlighting the power of single-cell analysis. Investigating the interaction between Tc17 cells and tumor cells in detail would also be an interesting direction for future studies. We demonstrated that Tc17 cells might originated from tissue-resident memory T cells and differentiate into the exhausted state via the analyses of clonotype and RNA velocity, depicting an alternative exhaustion

pathway in addition to the cytolytic-exhaustion path of T cells. Moreover, we identified TFs that are potential key regulators in the two exhaustion trajectories. Therefore, we hypotheses that these TFs can be manipulated, using CRISPR or RNAi, to induce the premature terminal-exhaustion of Tc17, and to intercept the exhaustion process of cytotoxic T cells. In summary, our results highlighted the therapeutic potential of targeting IL17⁺ T cells or its protumoral signaling (IL17/22/26) to treat IL17⁺ gastric cancer patients.

Collectively, out study illustrated a complex biological picture of GC, expound the associations between the cell subsets and tumor progression, and suggested some promising hints for tumor treatment. We hope that this cell atlas will survey as a valuable resource for GC research in the future.

One limitation of our study is the small number of patients. Therefore, the results of our analysis are exploratory, and should be further validated in large-scale scRNA-seq cohorts. Secondly, scRNA-seq lacks the crucial information of spatial distribution and chromatin accessibility of the various types of cells. In the future, we plan to apply spatial transcriptomics and scATAC-seq to dissect the positional relationship of these interacting cell types inferred by CellPhoneDB, and investigate the function of TFs in the remolding of cell states. Thirdly, our findings, such as the protumoral characteristics of TASCs, should be further verified and extended in patient-derived xenografts (PDXs), patient-derived organoids (PDOs) or other model systems. What's more, we will further investigate the underlying molecular mechanisms and regulatory pathways of cellular phenotypic remodeling in GC by genetically engineered mouse models (GEMMs) and other approaches.

## Methods
### Patients and samples
This study was approved by Independent Ethics Committee of the National Cancer Center/Cancer Hospital, Chinese Academy of Medical Science, and Peking Union Medical College, and all patients signed informed consent. Thirty gastric cancer patients who were pathologically diagnosed with GC were enrolled in this project, and they received none chemotherapy, radiation, or drug treatment before tumor resection. Detailed clinical information for these patients is provided in Supplementary Data 1. For patients GC03-GC10, their paired adjacent paratumor tissues were obtained during surgery. The adjacent paratumor tissues were taken more than 2 cm away from the matched tumor tissue. For patients GC06, GC07, GC08, and GC10, we also collected their peripheral blood prior to their surgical procedures. For patient GC08, we collected two spatial sites within one tumor due to the large tumor size.

### Sample collection, single-cell suspension processing, flow cytometry, and cell sorting
Briefly, fresh tissue samples were cut into small slices and enzymatically digested in the 10 ml RPMI-1640 medium containing 10% fetal bovine serum (FBS; GIBCO, Cat: 16000044), 1 mg/ml Collagenase type II (Gibco, Cat: 17101015), 1 mg/ml Collagenase type IV (Gibco, Cat: 17104019), 2 mg/ml Dispase II (Roche, Cat: 4942078001), 1 mg/

mL DNase I (Roche, Cat:10104159001), at 37 °C with 150 rpm rotation for 30 min. Following digestion, digested tissue pieces were passed through a 70-μm filter. The suspended cells were spun down at 400 g at 4 °C for 5 min and resuspended in ACK lysis buffer for 3 min on ice to remove red blood cells. After washing twice with 1x PBS (GIBCO, Cat: C10010500BT), the cell pellets were resuspended in sorting buffer (PBS containing 2% FBS). Peripheral blood mononuclear cells (PBMCs) were isolated using HISTOPAQUE-1077 (Sigma-Aldrich, Cat: 10771) solution according to the manufacturer instructions. After red blood cells were removed via the same procedure described above, PBMCs were also resuspended in the sorting buffer. All samples were stained for sorting at $1 \times 10^6$ cells per ml for 20 min on ice with Fixable Viability Dye eFluor™ 506 (eBioscience, Cat: 65-0866-18; 1:1000) for live-dead discrimination. For patients GC02 and GC08, tumors were stained with PerCP-Cy5.5 Mouse Anti-Human CD45 (BD Bioscience, Cat: 564105; 1:200) and PE Mouse Anti-Human CD3 (BD Bioscience, Cat: 555340; 1:200) antibodies to additionally isolate and enrich tumor-infiltrating T cells. Fluorescence-activated cell sorting (FACS) was performed on a FACSAria III instrument (BD Biosciences).

For comparing the proportion of MHC class II positive endothelium in tumor versus para-tumor tissue, single-cell suspensions of paired tumor and paratumor tissues from another nine patients were stained with Fixable Viability Dye eFluor™ 506 (1:1000), AF700 mouse anti-human CD45 (BioLegend, Cat: 304023; 1:200), PB mouse anti-EPCAM (BioLegend, Cat: 324217; 1:200), PE mouse anti-CD31 (BioLegend, Cat: 303105; 1:200) and FITC mouse anti-HLA- HLA-DR, DP, DQ (BioLegend, Cat: 361705; 1:200) antibodies.

## Single cell RNA-seq library preparation

The scRNA-Seq, scTCR-seq, and scBCR-seq libraries were prepared following the protocol provided by the 10X genomics Chromium Single Cell Immune Profiling Solution kit. Briefly, FACS-sorted cells were washed once with sorting buffer and resuspended in ice-cold PBS containing 1% FBS. The concentration of single cell suspensions was adjusted to 800–1200 cells/ul using a hemocytometer. 7000–14,000 cells were loaded in one channel, which resulted in a recovery of 4000–8000 cells. After partitioning cells into nanoliter-scale Gel Beads-in-emulsion (GEMs) using Chromium Single Cell 5′ Library & Gel Bead Kit (10× genomics, Cat: 1000006), reverse transcription (RT) was performed in a thermal cycler (Bio-Rad C1000 Touch). Then GEMs were broken, barcoded-cDNA was purified with Dynabeads MyOne SILANE and amplificated by polymerase chain reaction (PCR). The resulting amplified-cDNA was then used for 5′ gene expression library construction, TCR, and BCR enrichment. 50 ng of cDNA was used for fragmentation, end-repair, size-selection with SPRIselect beads, adapter ligation, and sample index PCR to construct 5′ gene expression library. Following two rounds of semi-nested PCR amplification using Chromium Single Cell V(D)J Enrichment Kit (10× genomics, Cat: 1000005, 1000016), 50 ng of enriched TCR/BCR product was used to construct scTCR/BCR-seq library. For GC03-tumor and GC10-paratumor tissues, two technical replicates of scRNA-seq libraries were processed to evaluate technical stability (Supplementary Fig. 1c). All libraries were sequenced on an Illumina Hiseq Xten with 150 bp paired-end reads.

## Single cell RNA-seq data processing

FASTQ reads of 10× scRNA sequencing data were processed with GRCh38 reference genome using Cell Ranger (version 3.1, 10× Genomics). The processed matrices of different batches were merged and the following analyses were done by scanpy (version 1.4.5)[61]. Cells with less than 400 UMI counts, less than 200 genes, or greater than 30% of mitochondrial RNA counts were filtered. Genes expressed by less than 3 cells were removed. The filtered expression matrix was normalized by the total number of UMIs per cell and was log2-transformed. Finally,

the filtered matrix contains 166,533 cells, 1620 genes per cell, and 5518 counts per cell, on average.

Scaled data of all cells were used for principal component analysis (PCA) using highly variable genes. The first 10 principal components and 500 neighbors were used for UMAP embedding with the first 2 principal components as UMAP initialization. Then, the Leiden algorithm was used to define clusters.

Different cell types were isolated and went through PCA, UMAP, and Leiden clustering independently for further analysis. For these analyses, the first 30 principal components were used and the numbers of neighbors ranged from 30 to 100, according to the cell numbers. Subclusters expressing contradictory markers (Supplementary Data 2) of known different cell types were removed as potential doublets. T cells with greater than 2 TRB or 2 TRA sequences were regarded as doublets. For T cells analysis, only cells with detected TRB sequences were used and the filter threshold of mitochondrial RNA counts was altered to 20%. In the remaining cells in of the T&NK cluster excluding αβ T cells lacking TCR information, we identified 4 NK clusters, 4 Gamma-delta (γδ) T clusters and 2 NKT clusters (Supplementary Fig. 12a–e). B cells, which make up a considerable part of immune cell compartment, were further grouped into 7 subsets (Supplementary Fig. 12f–j). Other subsets were discussed in detail in main text. Different cell subsets were named according to the expression of marker genes for clusters (Supplementary Data 3).

## Single-cell TCR-seq and BCR-seq data processing

10X genomics provides software for assembling V(D)J sequences and annotating consensus T cell receptor (TCR). Using the "Cell Ranger for V(D)J" pipeline, 78% of T cells annotated by transcriptomic data were assigned a TCR sequence. Overall, 4.8% of these T cells were assigned two TRB sequences and 8.6% were assigned two TRA sequences. Considering that the raw definition of clonotype grouping was only based on sequences in each sample, we combined clonotypes with identical TCR sequences among different samples of each patient. Finally, we detected 36,239 total clonotypes in all patients, most of which (30,980) were non-clonal clonotypes. The other clonotypes consisted of more than one cell and the largest clonotype size was 569 from patient GC10.

## Bulk sequencing data processing and single-cell variants extraction

FASTQ reads of WES data were aligned on GRCh38 reference and the somatic mutations were called by Strelka2[62] with paired tumor-normal samples. These mutations called from WES data were used as a reference to extract mutations in the scRNA-seq data using VarTrix tailored for 10X Genomics single-cell data. (The software is available at https://github.com/10xgenomics/vartrix). The expression quantification was performed with Salmon[63] using the FASTQ reads of RNA-seq data with GRCh38 reference.

## Batch correction for T cell analysis

In the part of the T-cell analysis, SCTransform from Seurat 3.1.0[64] was used to remove batch effect (variable.features.n = 10000, do.scale = TRUE), and the output data was used for PCA, UMAP, and Leiden clustering. For the evaluation of batch effects of T cells after the correction as well as of other cell types, see Supplementary Note 1. In addition, Supplementary Note 2 provides an evaluation of the robustness and necessity of T-cell clusters.

## Ratio of observation to expectation for tissue enrichment analysis

To evaluate the tissue preference of cell clusters, the observed cell number of each cell cluster from each tissue was divided by the expected cell number to get the ratio of observation to expectation (Ro/e) as described by Zhang et al.[30]. For a two-way table (cell clusters

for rows, tissues for columns) of a particular cell type, the expected cell number was calculated by (row total×column total)/$n$, where $n$ is the total observed cells in the table.

## Cellular detection rate and the combinations of different datasets

To calculate the cellular detection rate (CDR) of a particular gene in a certain cell type, we divided the number of cells expressing that gene by the total cell number in that cell type. In other words, CDR is equal to one minus the dropout rate.

To combine the DEGs from our scRNA-seq and our bulk RNA-seq for the same patient, we filtered the DEGs from bulk RNA-seq by adjusted $p$-value < 0.05 and |log2(FC)| > 1. For DEGs from scRNA-seq, we filtered them by adjusted $p$-value < 1e−10, |log2(FC)| > 1 and CDR outside epithelial cells < 5%. The intersection of remaining DEGs from the two datasets was shown in Fig. 2e.

To combine the correlation analysis from our scRNA-seq, our bulk RNA-seq, and the bulk RNA-seq from TCGA-STAD, we firstly selected genes with CDR outside epithelial cells < 5% and then calculated the Spearman's rank correlation between CDX2 and other genes with the three datasets independently. The correlation coefficients from the three datasets were multiplied together to get the combined correlation.

## Copy number variation (CNV) analysis

For each patient, epithelial cells excluding endocrine cells were considered as the putative tumor epithelium dataset. Other cells including immune cells and stromal cells were considered as the reference dataset. The initial CNV value of each single cell were estimated by infercnv (version 0.99.0) based on transcriptomic profiles as described by Puram et al.[12]. To quantificationally evaluate the CNV level of each single cell to identify malignant cells, we defined CNV_Score by calculating the mean squares of CNV values across the genome. The CNV results of WES (Supplementary Figs. 2a and 16) were generated by CNVkit with paired tumor and paratumor/blood samples.

## The calculation of tumor scores by expression patterns

We calculated malignant scores and non-malignant scores for epithelial cells using AddModuleScore function in Seurat[64], with the malignant gene set and the non-malignant gene set described by Zhang et al.[7]. These gene signatures were derived from the top 50 differential expression genes between paired tumor and normal tissue samples from the TCGA-STAD dataset (Supplementary Data 8). The tumor scores were calculated by subtracting the non-malignant scores from the malignant scores.

## Similarity analysis

Pearson correlation or Spearman's rank correlation was used to evaluate the similarity across cells, genes, or cell clusters. For cell clusters, the mean values of 1000 highly variable genes of the cells in each cluster were used to calculate the pairwise Pearson correlation. For cell-cell correlation, the first 50 principal components of each cell were used to calculate the pairwise Pearson correlation. For gene-gene correlation (co-expression analysis) in T cells, all the T cells with TCR information were used to calculate the pairwise Pearson correlation. As for the annotation bar on the top of Supplementary Fig. 7c, each gene was annotated to a T cell cluster as well as a tissue type with the highest mean expression among clusters or tissue types.

## Gene sets enrichment analysis

Gene sets of KEGG pathways and Gene Ontology were obtained from MSigDB[65]. Angiogenesis scores were calculated based on "angiogenesis (GO:0001525)". Cytolytic score, Anti-inflammatory score, M1 score and M2 score were calculated with gene sets of "Cytolytics effector

pathway", "Anti-inflammatory", "M1 Signature", and "M2 Suppressive Signature", respectively, from Azizi et al.[32].

For the scores of cell clusters shown in the heatmaps, the mean expression level of each cell cluster was used to calculate the score by GSVA (version 1.30.0) with default parameters[66]. For the score of a single cell shown in the boxplots, the average expression of the genes (z-score transformed) in the gene set was calculated.

## The activity of transcription factor regulon

The activity of transcription factor regulons was evaluated by SCENIC (version 1.0.1)[23]. In brief, regulons were detected by calculating the co-expression of TFs and genes, followed by motif analysis. AUCell score, ranged from 0 to 1, was then calculated by the algorithm for each cell to evaluate the activity level for each TF regulon. The regulon analyses were implemented independently for different main cell types.

## Inferring cell-cell communication

For systematic analysis of cell-cell interactions, we used CellPhoneDB (version 2.0)[67], a statistical algorithm for predicting cell-cell interaction networks from single-cell transcriptomic data. Ligand-receptor pairs were stored in CellPhoneDB and some immune-related pairs were downloaded from published literature[30]. In consideration of inter-tissue interactions, we divided each cell subtype into blood, paratumor or tumor sections and then remained sections based on the Ro/e value >1 as tissue-enriched subtypes. For each patient, we identified potential ligand-receptor pairs between two cell subtypes by measuring the expression of a receptor by one cell type and a ligand by another. The cell subtypes with fewer than 20 cells were excluded in each patient. Only ligands and receptors expressed in greater than 30% cells in any given subtypes were considered. To ensure high-confidence interactions, our pairwise subtypes analysis was performed by randomly permuting the subtype labels of all cells 1000 times. After permutations, statistical significance was assessed by the empirical $p$-value for each ligand-receptor pair between two cell subtypes. The interactions that were significant ($p$-value < 0.01) in at least one patient were counted when calculating the number of all possible interactions between the clusters.

The visualization of all networks was performed using Cytoscape (version 3.6.1)[68]. The nodes and edges each represent the cell subtypes and interactions in a network. The size of node is the total number of significant interactions with other cell subtypes, and the thickness of edges is the relative number of interactions between the connecting subtypes.

## RNA velocity and pseudotime analysis

The bam files generated by Cell Ranger were used to recount the spliced reads and unspliced reads using DropEst (version 0.8.6)[69]. The RNA velocity analysis was done with scvelo (version 0.1.25)[70] in python. In brief, after the gene selection and normalization, the first- and second-order moments were calculated with scv.pp.moments() function. The full splicing kinetics were recovered with scv.tl.recover_dynamics() function and the velocities were obtained with scv.tl.velocity() function in dynamical mode. The velocities were projected onto diffusion maps and visualized as streamlines with scv.pl.velocity_embedding_stream() function. The spliced vs. unspliced phase portraits of individual genes were visualized with scv.pl.velocity(). The pseudotimes of cells were obtain with scv.tl.recover_latent_time() function.

## Assessing the clonal migration between blood and tissues

To assess the migration potential of T cell clusters, we applied the Morisita−Horn similarity index to account for the number of shared clonotypes and the distribution of clone sizes between blood and solid tissue[71]. The Morisita−Horn similarity index is calculated by in each

cluster:

$$Index = \frac{2\sum\limits_{i=1}^{n} T_i B_i}{\sum\limits_{i=1}^{n}(T_i^2 + B_i^2)} \quad (1)$$

where $T_i = t_i/T_N$ and $B_i = b_i/B_N$, $b_i$ and $t_i$ are the clone sizes of the $i$th clonotype in blood and tumor tissue, and $B_N$ and $T_N$ are the total number of clone size of all clonotypes in blood and tumor tissue, respectively. There are over all n clonotypes in both tissues. The higher the index, the higher the migration potential between blood and tumor tissue.

### The usage of VDJ genes of T cell clusters

Through single-cell TCR sequencing, we distinguished the VDJ genes of each T cell. For each T cell cluster, the frequency of occurrence for VDJ genes was calculated. This VDJ-usage matrix was scaled with z-score before the PCA.

### Estimating cell type proportions in bulk RNA-seq data

We applied MuSiC (ver. 0.2.0)[27] to implement bulk tissue cell type deconvolution with scRNA-seq data. All defined subclusters in the scRNA-seq were used as reference. Subclusters with cell numbers larger than 3000 were down-sampled to 3000 cells to lighten the computation. Bulk RNA-seq data from TCGA-STAD were used for the deconvolution.

### Immunohistochemistry and multiplexed immunofluorescence staining

For immunohistochemistry, the 5-μm tissue sections of formalin fixed paraffin-embedded (FFPE) tissue were deparaffinized, and antigen was retrieved with sodium citrate, then stained with anti-FAP (abcam, Cat: ab207178; 1:250), anti-BMP1 (abcam, Cat: ab205394; 1:100), and anti-WNT5A antibodies (abcam, Cat: ab179824; 1:100). The images were captured with a Pannoramic scan (3DHISTECH). Multiplex staining of FFPE tissue was performed using the Opal 7-Color IHC kit (Akoya Biosciences, Cat: NEL811001KT) according to manufacturer's instruction. Briefly, the sections were blocked with 10% normal goat serum for 30 min after deparaffinization, rehydration, antigen retrieval, and endogenous peroxidase inactivation. Then, the sections were incubated with primary antibodies of different panel in a humidified chamber at 4 °C overnight, followed by horseradish peroxidase-conjugated secondary antibody incubation and tyramide signal amplification (TSA). The slides were microwave heat-treated after each TSA operation. Nuclei were stained with DAPI after all the antigens above had been labeled. The stained slides were imaged and scanned using the Vectra Quantitative Pathology Imaging Systems and analyzed by Phenochart Image Analysis Software (Akoya Biosciences) version 1.0.12. The primary antibodies and IHC metrics used in this paper were listed in Supplementary Data 7.

### Reporting summary

Further information on research design is available in the Nature Research Reporting Summary linked to this article.

## Data availability

The raw sequence data reported in this paper have been deposited in the Genome Sequence Archive in BIG Data Center, Beijing Institute of Genomics (BIG), Chinese Academy of Sciences. The raw sequence data are accessible at the following address (Access numbers: HRA000704). The processed expression matrices and cell annotations have been deposited into Open Archive for Miscellaneous Data (OMIX) database with accession ID: OMIX001073. All other relevant data supporting the key findings of this study are available within the article and its Supplementary Information files. Source data are provided with this paper.

## Code availability

Example scripts to process and analyze data are available at https://github.com/Lan-lab/sc-GC. Detailed information will be available from the corresponding authors upon reasonable request.

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

## Acknowledgements

We thank Yan Liu for technical support in fluorescence-activated cell sorting. This work was partially supported by the grants (Grant No. 81972680 to X.L.) from National Natural Science Foundation of China, the grants (Grant No. 61020100119 to X.L.) from Tsinghua University-Peking University Jointed Center for Life Science, the grants (Grant No. 042021011 to X.L.) from National Thousand Young Talents Program of China, the grants (Grant No. 2019M660683 to X.S.) from China Postdoctoral Science Foundation, and a start-up fund from Tsinghua University-Peking University Joined Center for Life Science.

## Author contributions

K.S., Y.T. and X.L. designed experiments. K.S, F.M, Y.L., X.S., P.J., W.K., L.J., J.X., and H.H. performed the experiments. R.X., K.S, N.Y. and X.L. analyzed sequencing data. K.S., R.X., N.Y. and X.L. wrote the manuscript, with all authors contributing to writing and providing feedback.

## Competing interests

The authors declare no competing interests.
