## [Peer Review File · Nature Communications]

scRNA-seq of gastric tumor shows complex intercellular interaction with an alternative T cell exhaustion trajectoryREVIEWER COMMENTS

Reviewer #1 (Remarks to the Author):

The authors generated a valuable scRNA seq dataset of 166,533 cells from 10 gastric cancer (GC) patients with matched adjacent normal tissues and peripheral blood, and co-presented paired TCR and BCR (T/B cell receptor) sequencing data to investigate the state transitions within different T/B cell subtypes. Inferred CNV scores divided the epithelial cells into tumor, tumor-like and normal. They described significant intertumoral and intratumoral heterogeneity seen in the tumor epithelial cell compartments. For stromal cell analysis, they found that the stromal cells in the tumor tissue undergone a significant transformation and showed extensive protumoral features. Tumor-associated stromal cells (TASCs), comprising of endothelial, fibroblast and smooth muscle cell, were implicated in epithelial mesenchymal transition and angiogenesis, and negatively correlated with survival. For myeloid cell populations, the authors analyzed to find subsets representing tumor-associated macrophages which were enriched by lipid and lysosome-related genes. For dendritic cells (DCs), they found a possible origin of LAMP3+ DCs, a migrating population, from cDC2_CD1C by RNA velocity analysis. They presented two trajectories that suggest Tc17 (IL-17+ CD8+ T) cells originate from tissue-resident memory T cells and can differentiate into exhausted T cells. Lastly, they present a hypothesis that IL17+ Tc17 and Th17 cells may promote tumor progression through IL17, IL22, and IL26 signaling.

The primary message from this paper is that extensive remodeling of TMEs occur in GC, particularly in the stromal cell components. The gene signatures of TME remodeling were prognostic. The secondary message is the data itself, as the authors claimed, which will be useful for other researchers to dissect the complex pathophysiology of GC. While the efforts of generating such a huge dataset and analyzing it must be prized, the major claims of this paper – TME remodeling and immune cell activation/exhaustion seen in scRNAseq level itself are not novel findings in this field.

- Davidson, Sarah et al. "Single-Cell RNA Sequencing Reveals a Dynamic Stromal Niche That Supports Tumor Growth." *Cell reports* vol. 31,7 (2020): 107628.

doi:10.1016/j.celrep.2020.107628

- Sathe, Anuja et al. "Single-Cell Genomic Characterization Reveals the Cellular Reprogramming of the Gastric Tumor Microenvironment." *Clinical cancer research : an official journal of the American Association for Cancer Research* vol. 26,11 (2020): 2640-2653. doi:10.1158/1078-0432.CCR-19-3231

- Wang, Bin et al. "Comprehensive analysis of metastatic gastric cancer tumour cells using single-cell RNA-seq." *Scientific reports* vol. 11,1 1141. 13 Jan. 2021, doi:10.1038/s41598-020-80881-2

- Wang, Ruiping et al. "Single-cell dissection of intratumoral heterogeneity and lineage diversity in metastatic gastric adenocarcinoma." *Nature medicine* vol. 27,1 (2021): 141-151. doi:10.1038/s41591-020-1125-8

- Zhang, Min et al. "Dissecting transcriptional heterogeneity in primary gastric adenocarcinoma by single cell RNA sequencing." *Gut* vol. 70,3 (2021): 464-475. doi:10.1136/gutjnl-2019-320368

- Zhang, Peng et al. "Dissecting the Single-Cell Transcriptome Network Underlying Gastric Premalignant Lesions and Early Gastric Cancer." *Cell reports* vol. 27,6 (2019): 1934-1947.e5. doi:10.1016/j.celrep.2019.04.052

Yet, the authors present very interesting findings, such as lipid and lysosome-related genes enriched myeloid cell populations (possibly the TAMs), LAMP3+ DCs cell of origin and Tc17 (IL-17+ CD8+ T) cells exhaustion pathways, and IL17, IL22, IL26 signaling role in GC progression. The limitation here is that they present a single line of evidence, for multiple associations rather and cause/result relationships. If the authors have access to the tumor tissue, they can perform immunohistochemistry of the suggested TASC markers and present them as secondary line of evidences. Or, more precisely, they can use a model system to prove their hypothesis.

The authors state that cancer hallmark pathway analysis revealed that both angiogenesis and epithelial-mesenchymal transition (EMT) pathways, the key signatures of tumor progression, were significantly upregulated in TASCs (Fig. 2f and Extended Data Fig. 3g, h). However, the cells are claimed to be endothelial cells, fibroblasts, and smooth muscle cells, Therefore, it is unclear what "angiogenesis" and "EMT" pathway enrichments represent for those non-epithelial cell populations. They argue that NFKB1 and SOX4 transcription factor (TF)s might commonly regulate the genes in TASCs, yet again it is unclear why such TFs should act in similar manner in different cell populations. If it is NFKB1, one can hypothesize that IL-6 or other cytokines activate NFKB in TASCs.

Reviewer #2 (Remarks to the Author):

In the submitted manuscript, Sun et al. analyzed the single cell RNA sequencing data of 166,533 cells collected from 10 treatment-naïve gastric cancer patients, with tumor tissue (n=10), matched normal tissues (n=8), and blood (n=5). They characterized the epithelial, immune and stromal compartments within the tumor microenvironment and their cellular interactions. The TME has not been well studied in gastric cancer and this study discovered that IL17+ cells may promote tumor progression and nominated it as a therapeutic target to treat gastric cancer. The manuscript suffers from major limitations. The paper is in general descriptive and the number of samples studied is small. Importantly, the analysis is overall superficial, with limited insights adding to current knowledge.

Major comments:

1. Gastric cancer is histologically heterogenous, according to Lauren's classification, it can be classified into 3 main types, intestinal, diffuse, and mixed type, and the diffuse type can be further classified into signet ring cell carcinoma based on the presence of signet ring cells. Among the 10 patients characterized by this study, 2 were intestinal type, 2 were diffuse type, and 6 were mixed type. No information is provided on the signet ring cells. Also, no information is provided about the stages and sites of the tumors, or the MSI status. The cohort size is very small given the high degree of heterogeneity (genomic, epigenomics, transcriptomic, histopathological, etc.) However, the analysis results in this study were not correlated with any of these clinical and histopathological variables mentioned above.

2. The definition of tumor cells is problematic. In this study, tumor cells were defined solely based on CNV_score inferred from gene expression profiles; however, it is well known that a large fraction of primary gastric adenocarcinomas is genomically stable (GS) (Nature 2014, PMID: 25079317). Likewise, tissue enrichment analysis (Ro/e) can not be used to identify tumor cells, although theoretically, tumor tissues should contain more tumor cells than the adjacent normal tissues. However, due to sampling bias, tumor tissues may not always contain tumor cells and tumor cells can also found in adjacent normal tissues due to contamination or sampling issue. In addition, the definition of tumor-like clusters is confusing and a bit misleading. It is unclear how these tumor-like cells are defined. For example, cells of tumor-like cluster C4 mainly expressed normal epithelial lineage markers (PGA3, PGC, MUC6) and did not express marker genes for gastric cancer (Fig. 1i). Tumor-like cluster C13 mainly express marker gene of chief cells (PGA3, PGC) and parietal cells (ATP4A, ATP4B). The genes such as OLFM4, LGR5, and CD44, KRT7 can also express in normal epithelial cells. Clearly, the tumor like cells are transcriptionally heterogenous, as some cells are clustered closely with normal epithelial cells while some others are clustered closer to tumor cells. Analysis of tumor cells has to be significantly improved.

3. The analysis performed on tumor cells is very superficial. The authors simply defined the tumor cells clusters but without performed any further profiling on tumor cells. The

insights derived from this part is very limited. Also, a clear limit is that, although ten tumors were sequenced, the epithelial and tumor cells were mainly derived from 4 tumors: GC08, GC10, GC07, GC02.

4. TASCs cells are interesting and the authors showed that the TASC signatures were associated with a worse survival in TCGA-STAD cohort. However, this survival difference may not be attributed to the presence of TASCs in the tumor microenvironment, because it is very possible that tumor cells and other cells may also express the signature genes of TASCs as the TASCs signature genes may not be exclusive to TASCs and the authors would need to provide more evidence on this. Also, potentially confounding factors were not considered in their analysis. Similarly, DEGs of Macrophage_APOE could express in cells other than macrophages and TCGA data is not directly supportive of their biological significance and functional studies are needed.

5. The TFs analysis in this study was very descriptive, the results were kind of isolated, and the significance of such analysis is not evident.

6. The author stated that the DC_LAMP3 is less-described cluster, which is incorrect. This cell population has been well described by Dr. Zemin Zhang's group (Zhang et al. Cell 2019; Cheng et al. Cell 2021) and other groups. Also, the differentiation origin of LAMP3+ DC from cDC2 has been well described by Cheng et al. (Cell, 2021, PMID: 33545035), and therefore is not novel.

7. The authors identified 21 T cell clusters (Fig. 4a-b), some of these clusters are very small and not clearly separated from other cells. Also, the signature genes showed in 4b are largely shared among the CD4/CD8 cell subclusters. This raises concerns about the reproducibility of their clustering analysis and validations of the computational method are therefore warranted. It would be useful and necessary to demonstrate how robust the clustering is and the reproducibility of their results with additional clustering approaches (k-mean or hierarchical clustering based) and down sampling the cells (e.g. 25%, 50%).

8. The proportion of cells expressing the marker genes IL17A and IL23R in the two cell clusters CD4_C6_IL17A and CD8_C8_IL17A was very very low (Fig. 4b based on the size of the circles). This could possibly be due to the high gene dropout rate of 10x sequencing. Nevertheless, defining cells based on expression of these genes are not proper and has to be validated by additional approaches. The multicolor IHC showed in Fig. 4d is not convincing as only a several of such cells are shown in the images.

9. Cell-cell communication analysis is potential interesting, but this part is purely descriptive without any functional evaluation. At least, correlation analysis should be performed to check whether the frequencies of those interacting cells are significantly correlated. The authors pooled cells from all samples and patients for Cell-cell communication analysis without take into consideration that these "interacting" cells (shown on the heatmap) may not even present in the same samples. For the methodology, it is unclear how the data are filtered, and whether those rare cell subsets with only few cells were excluded and the cutoff applied.

10. Line 383-390: The authors described VDJ-gene usage, however, it is unclear how are the differences in VDJ-gene usages in T cell clusters related to the pathogenesis of gastric cancer.

11. Tc17 cells have to be functionally defined before further analyzing their developmental trajectories. What are the marker genes of these cells and what are their roles in anti-cancer immune response? T cell exhaustion trajectories have been well described in many other studies, and therefore it is unclear how results from current study can add to existing knowledge.

12. There is no description of doublets removal in the Methods, it is therefore not sure

that doublets were carefully removed.

13. No statistical assessment of the batch effects is performed, an evaluation of the batch effects (whether it was present and how significant it was) were not described or in the Methods, or demonstrated in the main figures or suppl figures.

14. Potentially cofounding factors were not considered in the survival analysis.

Minor comments:

15. Line: 141-143: it is difficult to read the two figure panels as the patient IDs were not labelled on Fig. 1i and the genes mentioned here (FABP1, MUC5AC, MUC5B) were not labelled on Fig. 1j.

16. Line 167: the author stated that Fib_1 expressed IL11 and IL24 while the expression levels of these two genes were very low and not easily visible in Fig. 2C.

17. Figure 3b, the color key for standardized expression levels was not generated properly.

18. Line 206: no references added for "other studies".

19. Figure 3c, the pathway name on the right bottom was not displayed properly.

20. It is not clear how the endocrine cells and smooth muscle cells were defined as no commonly used canonical markers of these cells are seen in their figure 1c or extended data figure 1d. The smooth muscle cells and fibroblasts had many overlapping genes and are not clearly distinguishable from each other on their figures.

21. Are GC03 and GC10 technical duplicates or biological duplicates? Were the libraries made using the same tissue?

Reviewer #3 (Remarks to the Author):

Sun et al. have performed a detailed analysis of cells infiltrating the gastric cancer (GC) tumor microenvironment, creating an atlas of >166,000 cells from 10 GC patients. Such cells were studied by scRNAseq, coupled with TCR sequencing. Among other findings, they report that tumor-associated macrophages and dendritic cells expressing LAMP3 were able to mediate T cell activity. Furthermore, they describe the role and function of IL-17 producing CD8+ T cells, a possible novel therapeutic target. The paper is interesting and technically well done; the statistical analysis and the presentation of data are adequate.

Comments:

1) Lines 603-605: "Cells expressing contradictory markers of known different cell types were removed as potential doublets". It would be better to indicate the markers used to identify the doublets and the number of excluded cells.

2) Lines 625-629: "... SCTransform .. was used to remove a batch effect... ". The figures 4a and 4c report that some clusters are tissue-dependent, like CD8-C1, CD4-C1, CD4-C2. Naïve CD8 (CD8-C1) and naïve CD4 (CD4-C1) are close in the UMAP space, and it has to be excluded that was due to some source of technical variability. This variability can also cause the finding of other clusters. Have you tried to regress out other source of

variations, like the mitochondrial gene percentages (that likely could be different among peripheral cells and TIL)? Additionally, some supplementary UMAP plot colored by the number of gene (and mitochondrial gene) expressed by each cluster could help in understanding the results of clustering process.

3) Figure 4: The clusters CD4_C2_LTB and CD4_C3_SLC2A3 designated as memory likely have to express also CREB or S1004A. If possible, highlining these transcripts would be indicative. Then, for the CD4_C4_CD69 cluster we recommend reporting the expression of CXCR6 and ANXA1, considering that cells were indicated as TRM (see: Szabo, Peter A., et al. "Single-cell transcriptomics of human T cells reveals tissue and activation signatures in health and disease." *Nature Communications* 10.1 (2019): 1-16).

4) Line 373: "We observed that CD8_C2(Effector), CD8_C3 (Cytotoxic), CD8_C4 (Effector memory), and CD8_C10 (MAIT) clusters have both higher proportions of clonal cells and proportions of clonal cells with shared TCR between blood and solid tissue (Fig. 6c)". Maybe using the percentage of overlapping TCR is not the optimal metrics to measure the shared clonotype among blood and solid tissue (or patient/cluster). The "Morisita similarity index", that also considers the size of the population, could be useful for this purpose.

5) Line 398: "we thus hypothesized two possible trajectories for T cell state transition (Fig. 6i)". This could be an interesting finding, but some additional plot showing the patient/sample effect would be indicative, and would greatly clarify if this observation is affected by the patient-effect rather than by the biological process.

Dr. Andrea Cossarizza and Dr. Domenico Lo Tartaro

Index

A brief summary of the revision.....	1
Reviewer #1 (Remarks to the Author):.....	2
Reviewer #2 (Remarks to the Author):.....	10
Reviewer #3 (Remarks to the Author):.....	35

A brief summary of the revision

We are very grateful to Dr. Andrea Cossarizza, Dr. Domenico Lo Tartaro and the two anonymous reviewers for their constructive criticism and comments on our study. In the revised manuscript, we made substantial improvements in the computational analysis and provided further experimental validations to address the reviewers' concerns.

Firstly, to provide information from additional sources to support our conclusion, we added whole-exome sequencing (WES) and bulk RNA-seq for the same samples and analyzed the Stomach Adenocarcinoma (STAD) dataset from The Cancer Genome Atlas (TCGA) as well as the gastrointestinal data from The Genotype-Tissue Expression (GTEx) dataset.

Secondly, we completely rewrote the parts regarding tumor cells and stromal cells with an emphasis on the biological functions and clinical relevance of the findings (e.g., tumor sites, intestinal metaplasia, Wnt signaling, MHC II complex, and angiogenesis) and added a new figure as Figure 2.

Thirdly, we validated the computational predictions of regulatory TFs with in vitro experiments (Fig. 4g). Moreover, we obtained new clinical samples and performed flow cytometry and multicolor immunohistochemistry (IHC) staining to further support our conclusions.

Last but not least, we improved a number of computational analyses according to the suggestions from the reviewers. Besides, we added a systematic evaluation of batch effects in Supplementary Note 1 as well as an evaluation of the robustness and necessity of T-cell clusters in Supplementary Note 2.

Point-by-point responses to the reviewers' comments

Reviewer #1 (Remarks to the Author):

The authors generated a valuable scRNA seq dataset of 166,533 cells from 10 gastric cancer (GC) patients with matched adjacent normal tissues and peripheral blood, and co-presented paired TCR and BCR (T/B cell receptor) sequencing data to investigate the state transitions within different T/B cell subtypes. Inferred CNV scores divided the epithelial cells into tumor, tumor-like and normal. They described significant intertumoral and intratumoral heterogeneity seen in the tumor epithelial cell compartments. For stromal cell analysis, they found that the stromal cells in the tumor tissue undergone a significant transformation and showed extensive protumoral features. Tumor-associated stromal cells (TASCs), comprising of endothelial, fibroblast and smooth muscle cell, were implicated in epithelial mesenchymal transition and angiogenesis, and negatively correlated with survival. For myeloid cell populations, the authors analyzed to find subsets representing tumor-associated macrophages which were enriched by lipid and lysosome-related genes. For dendritic cells (DCs), they found a possible origin of LAMP3+ DCs, a migrating population, from cDC2_CD1C by RNA velocity analysis. They presented two trajectories that suggest Tc17 (IL-17+ CD8+ T) cells originate from tissue-resident memory T cells and can differentiate into exhausted T cells. Lastly, they present a hypothesis that IL17+ Tc17 and Th17 cells may promote tumor progression through IL17, IL22, and IL26 signaling. The primary message from this paper is that extensive remodeling of TMEs occur in GC, particularly in the stromal cell components. The gene signatures of TME remodeling were prognostic. The secondary message is the data itself, as the authors claimed, which will be useful for other researchers to dissect the complex pathophysiology of GC. While the efforts of generating such a huge dataset and analyzing it must be prized, the major claims of this paper – TME remodeling and immune cell activation/exhaustion seen in scRNAseq level itself are not novel findings in this field.

- Davidson, Sarah et al. "Single-Cell RNA Sequencing Reveals a Dynamic Stromal Niche That Supports Tumor Growth." *Cell reports* vol. 31,7 (2020): 107628. doi:

[10.1016/j.celrep.2020.107628](https://doi.org/10.1016/j.celrep.2020.107628)

- Sathe, Anuja et al. "Single-Cell Genomic Characterization Reveals the Cellular Reprogramming of the Gastric Tumor Microenvironment." *Clinical cancer research: an official journal of the American Association for Cancer Research* vol. 26,11 (2020): 2640-2653. doi: 10.1158/1078-0432.CCR-19-3231
- Wang, Bin et al. "Comprehensive analysis of metastatic gastric cancer tumour cells using single-cell RNA-seq." *Scientific reports* vol. 11,1 1141. 13 Jan. 2021, doi:10.1038/s41598-020-80881-2
- Wang, Ruiping et al. "Single-cell dissection of intratumoral heterogeneity and lineage diversity in metastatic gastric adenocarcinoma." *Nature medicine* vol. 27,1 (2021): 141-151. doi:10.1038/s41591-020-1125-8
- Zhang, Min et al. "Dissecting transcriptional heterogeneity in primary gastric adenocarcinoma by single cell RNA sequencing." *Gut* vol. 70,3 (2021): 464-475. doi:10.1136/gutjnl-2019-320368
- Zhang, Peng et al. "Dissecting the Single-Cell Transcriptome Network Underlying Gastric Premalignant Lesions and Early Gastric Cancer." *Cell reports* vol. 27,6 (2019): 1934-1947.e5. doi:10.1016/j.celrep.2019.04.052

Reply:

We thank the reviewer for the suggestions. We have summarized the results and conclusions of previous studies applying single-cell RNA-seq in gastric cancer in the introduction section of the revised manuscript as follows (Page 3, lines 48-54):

"Recently, Single-cell RNA-sequencing (scRNA-seq) has been successfully used to decipher the ecosystems of GC, to dissect and uncover the underlying tumor biology of interest⁵⁻¹⁰. For example, Wang *et al.* and Zhang *et al.*, revealed the transcriptional heterogeneity and lineage diversity in primary and metastatic gastric adenocarcinoma, and provided signature genes for diagnosis and prognosis^{6,7}. Zhang *et al.* and Yin *et al.*, delineated the vast cellular phenotypic remodeling during GC occurrence and development, and also identified makers for early GC detection^{5,9}"

Yet, the authors present very interesting findings, such as lipid and lysosome-related genes

enriched myeloid cell populations (possibly the TAMs), LAMP3+ DCs cell of origin and Tc17 (IL-17+ CD8+ T) cells exhaustion pathways, and IL17, IL22, IL26 signaling role in GC progression. The limitation here is that they present a single line of evidence, for multiple associations rather and cause/result relationships. If the authors have access to the tumor tissue, they can perform immunohistochemistry of the suggested TASC markers and present them as secondary line of evidences. Or, more precisely, they can use a model system to prove their hypothesis.

Reply:

We thank the reviewer for the positive comment and for pointing out the limitations. To support our findings with additional lines of evidences, we have performed, 1. multicolor immunohistochemistry (IHC) staining to detect CD31⁺ HLA-DR⁺ endothelial cells (Fig. R1a), CD31⁺FAP⁺ tumor-associated endothelial cells (Fig. R1b), PDGFRA⁺FAP⁺ tumor-associated fibroblasts (Fig. R1c), and CD68⁺APOE⁺/CD68⁺THBS1⁺ macrophages (Fig. R1h). 2. flow cytometry on nine additional GC patient samples, which showed the fraction of MHC class II⁺ endothelial cells in paratumors was higher than that in tumors (Fig. R1g), consistent with the scRNA-seq data. 3. IHC to detect markers highly expressed in TASCs, including FAP, BMP1 and WNT5A (Fig. R1d-f). We added these results in the revised manuscript as follows,

“Notably, we found endothelial cells in stomach expressed major histocompatibility complex (MHC) class II genes such as *HLA-DRA* and *HLA-DRB5* (Fig. 3c), which was confirmed by multicolor immunohistochemistry (IHC) staining on tumor sections from GC patients (Fig. 3d). Moreover, we found that Endo_1 featured downregulated MHC class II genes (Fig. 3f), indicating that the intrinsic antigen presentation function of Endo_1 was limited. We further performed flow cytometry on nine additional GC patient samples, which showed the fraction of MHC class II⁺ endothelium in paratumors was higher than that in tumors ($p < 0.001$, Student’s paired t test) (Fig. 3g and Supplementary Fig. 3c), consistent with the observation in scRNA-seq data.

Both Endo_1 and Fib_1 expressed fibroblast activation protein (*FAP*), a classical cancer-associated fibroblast (CAF) marker. Similarly, we performed multicolor IHC staining to validate the presence of FAP⁺ fibroblast and endothelial cells in tumors (Fig. 3e and Supplementary Fig. 3d). Fib_1 also expressed others CAF markers, such as *MMP3* and *MMP11*, and inflammation-associated fibroblast markers (*IL11*, *IL24*) that promote carcinogenesis²⁴ (Fig. 3c and Supplementary Fig. 3g). Genes in the Wnt signaling pathway such as, *WNT2* and *WNT5A*, were upregulated in tumor fibroblasts while *SFRP1*, an inhibitor of Wnt signaling was downregulated (Fig. 3i). These genes also showed similar expression patterns in TCGA-STAD dataset (Fig. 3j). Note that these three genes were expressed almost exclusively by fibroblasts (Supplementary Fig. 3e).

Besides, Fib_1 cells exhibited upregulation of the *TWIST1-PRRX1-TNC* positive feedback pathway, which is known to promote the activation and expansion of CAFs in the TME²⁵. Meanwhile, bone morphogenetic protein 1 (*BMP1*) and *ANGPT2*, which respectively facilitate tumor growth and angiogenesis, were expressed at significantly higher levels in SMC_1. IHC results validated that the protein expression of FAP, BMP1, WNT5A were upregulated in tumor (Fig. 3h). Hereafter, we defined cells in the three tumor-enriched cell clusters, Endo_1, Fib_1, and SMC_1 as tumor-associated stromal cells (TASCs).” (Page 9-11, lines 176-200), and “We further performed multicolor IHC staining to validate the presence of the two distinct macrophages (Supplementary Fig. 5c).” (Page 12, lines 231-233).

In the future, we will continue to verify our hypothesis on the function of TASCs in tumor progression using patient-derived xenografts (PDXs) and genetically engineered mouse models.

Figure R1. (a-c) Multicolor IHC staining with anti-CD31 and HLA-DR antibodies showing HLA-DR⁺ endothelial cells (a), anti-CD31 and FAP antibodies showing FAP⁺ endothelial cells (b), and anti-PDGFR α and FAP antibodies showing FAP⁺ fibroblasts (c). The scale bar represents 20 μ m. **(d-f)** IHC staining of FAP, BMP1 and WNT5A on formalin-fixed and paraffin-embedded slides for the independent biospecimens. The scale bar represents 100 μ m. **(g)** Dot plot showing the higher proportion of MHC class II⁺ endothelial cells in paratumors than tumors. *** $p < 0.001$ (Paired t-test). **(h)** Multicolor IHC staining with anti-CD68, APOE and THBS1 antibodies. The red and green arrows indicate CD68⁺APOE⁺ and CD68⁺THBS1⁺ macrophage, respectively. The scale bar represents 50 μ m.

The authors state that cancer hallmark pathway analysis revealed that both angiogenesis and epithelial-mesenchymal transition (EMT) pathways, the key signatures of tumor progression, were significantly upregulated in TASCs (Fig. 2f and Extended Data Fig. 3g, h). However, the cells are claimed to be endothelial cells, fibroblasts, and smooth muscle cells, Therefore, it is unclear what “angiogenesis” and “EMT” pathway enrichments represent for those non-epithelial cell populations. They argue that NFKB1 and SOX4 transcription factor (TF)s might commonly regulate the genes in TASCs, yet again it is

unclear why such TFs should act in similar manner in different cell populations. If it is *NFKB1*, one can hypothesize that IL-6 or other cytokines activate *NFKB* in TASCs.

Reply:

We apologize for not being clear in this part of the analysis. Both cancer cells and stromal cells can secrete factors that stimulate angiogenesis, and hence, facilitate the development of cancer. The enrichment of angiogenesis in TASCs indicates that stromal cells may help establish an appropriate microenvironment for angiogenesis in GC by releasing various kind of factors. By calculating the angiogenesis score in epithelial and stromal cell clusters, we found that signature of angiogenesis is more pronounced in stromal cells than epithelial cell in general in GC (Fig. R2). Moreover, we found TASCs, including Endo_1, Fib_1 and SMC_1, had higher angiogenesis score than stromal cells enriched in paratumor tissues.

Figure R2. Violin plot showing the angiogenesis score in epithelial cell and stroma cell clusters. Normal.epi and IM cells represent normal epithelial cells and intestinal metaplasia cells, respectively. SMC_1, Endo_1 and SMC_1 are TASCs. SMC_2, Fib_others and Endo_others are enriched in paratumor tissues.

By ligand-receptor analysis, we found that Endo_1 demonstrated strong activations of tumor necrosis factor (*TNF*) signaling, vascular endothelial growth factor (*VEGF*) signaling, platelet derived growth factor (*PDGF*) signaling, placental growth factor (*PGF*) signaling and Notch signaling (Fig. 8a, c), which are important signaling pathways participating in angiogenesis. Notably, Endo_1, Fib_1 and SMC_1 were the main supplier for these factors

or cytokines in these pathways (Fig. 8a, c). This analysis further explains the upregulation of angiogenesis in TASCs and perhaps also in the tumor cells. We have added this analysis in the revised manuscript (Page 22, lines 455-458).

Epithelial-mesenchymal transition (EMT) is an evolutionarily conserved developmental program that enables carcinoma cells to replace their epithelial features with mesenchymal features. Therefore, as you mentioned, it is hard to interpret the upregulated EMT pathway in TASCs. We have deleted this part in the revised manuscript. We apologize for the misleading statements.

The activation of NF- κ B pathway can be triggered by proinflammatory cytokines such as TNF α and IL-1 (Karin, 2006; Taniguchi and Karin, 2018; Xia et al., 2014). In the part of cell-cell communication, we found Endo_1, Fib_1 and SMC_1 can interact with *OSM*, *IL6*, *IL1B*, and *TNF*, which are classical inflammatory cytokines activating NF- κ B and the key activators of stromal cells (Shi et al., 2017). This analysis suggests similar signaling pathways, related to these cytokines, are involved in the activation of the three tumor-associated stromal cell clusters, which may account for the upregulation of regulon score of *NFKB1* in TASCs to some extent (Fig. 3m). It is of interest that NF- κ B pathway is also involved in the activation of genes concerning angiogenesis (Karin, 2006; Taniguchi and Karin, 2018; Xia et al., 2014), and angiogenesis pathway was significantly upregulated in TASCs (Fig. 3k). Maybe, angiogenesis pathway in TASCs was regulated by *NFKB1*. We have added these analyses in the revised manuscript (Page 23, lines 463-466 and Page 11, lines 214-216).

We acknowledge that the causes of NF- κ B pathway activation are very complex. Our analysis can't fully explain the upregulation of *NFKB1* in TASCs. Whether *NFKB1* and *SOX4* play a role in TASCs should be further verified by Western Blot, Chip-seq, ATAC-seq and other experiments. Likewise, whether *OSM*, *IL6*, *IL1B*, and *TNF* can trigger the activation of NF- κ B signaling pathway in TASCs of gastric cancer should be further verified by other experiments. For example, we can stimulate the TASCs isolated from gastric tumor with *OSM*, *IL6*, *IL1B*, and *TNF*, then detect the expression of *NFKB1* in TASCs by qPCR and Western Blot. In the future, we will perform the experiments described above to

investigate the role of these TFs in TASCs in more detail. Our work aims to illustrate the complex biological ecosystem of GC by using scRNA-seq data. We recognized our limitations and discussed this in the discussion section of the article in the revised manuscript (Page 26, lines 535-545).

References

- Karin, M. (2006). Nuclear factor- κ B in cancer development and progression. *Nature* 441, 431-436.
- Shi, Y., Du, L., Lin, L., and Wang, Y. (2017). Tumour-associated mesenchymal stem/stromal cells: emerging therapeutic targets. *Nature Reviews Drug Discovery* 16, 35-52.
- Taniguchi, K., and Karin, M. (2018). NF- κ B, inflammation, immunity and cancer: coming of age. *Nature Reviews Immunology* 18, 309-324.
- Xia, Y., Shen, S., and Verma, I.M. (2014). NF- κ B, an active player in human cancers. *Cancer immunology research* 2, 823-830.

Reviewer #2 (Remarks to the Author):

In the submitted manuscript, Sun et al. analyzed the single cell RNA sequencing data of 166,533 cells collected from 10 treatment-naïve gastric cancer patients, with tumor tissue (n=10), matched normal tissues (n=8), and blood (n=5). They characterized the epithelial, immune and stromal compartments within the tumor microenvironment and their cellular interactions. The TME has not been well studied in gastric cancer and this study discovered that IL17+ cells may promote tumor progression and nominated it as a therapeutic target to treat gastric cancer. The manuscript suffers from major limitations. The paper is in general descriptive and the number of samples studied is small. Importantly, the analysis is overall superficial, with limited insights adding to current knowledge.

Reply:

Thank you for pointing out our limitations. We have made substantial changes in the revised manuscript to address the reviewer's concerns. Specifically, we added bulk RNA-seq and whole-exome sequencing (WES) of the same tissues to facilitate the analysis of epithelial cells and the identification of tumor cell. We also analyzed the TCGA-STAD and Genotype-Tissue Expression (GTEx) datasets to ensure our results can be observed in large cohorts. Moreover, we made effort to add more analysis and provide biological interpretations of the computational results. Furthermore, we performed experimental validations using new clinical samples to support our findings.

Major comments:

1. Gastric cancer is histologically heterogenous, according to Lauren's classification, it can be classified into 3 main types, intestinal, diffuse, and mixed type, and the diffuse type can be further classified into signet ring cell carcinoma based on the presence of signet ring cells. Among the 10 patients characterized by this study, 2 were intestinal type, 2 were diffuse type, and 6 were mixed type. No information is provided on the signet ring cells. Also, no information is provided about the stages and sites of the tumors, or the MSI status. The cohort size is very small given the high degree of heterogeneity (genomic, epigenomics, transcriptomic, histopathological, etc.) However, the analysis results in this

study were not correlated with any of these clinical and histopathological variables mentioned above.

Reply:

We apologize for the limited information shown in Fig. S1 and for the misleading names used in the supplementary data. The detailed clinical information of patients is now provided in the file Supplementary Data 1 (named Supplementary Table 1 in the first submission) with information including stages, sites, tumor size, TNM classifications. We applied MSIsensor (Niu, B. et al., 2014) on the WES data to infer the MSI status and added the results in Fig. S2g.

As the reviewer suggested, we attempted to link the analysis results to clinical variables in the revised manuscript. We evaluated the degree of intestinal metaplasia at transcriptional level (Fig. R3a). Then, we analyzed genes associated with intestinal metaplasia and predicted potential upstream regulators of *CDX2* (Fig. R3b), an essential TF involved in intestinal metaplasia. Furthermore, we observed abnormal expressions of Wnt-related genes in GC08 tumor cells that made the tumor cells resemble intestinal stem cells (Fig. R3c). By investigating the mutation information from WES, we hypothesized that APC mutation was associated with the changes in the expression of *CDX2* and Wnt-related genes and then supported the hypothesis with the result from TCGA-STAD dataset (Fig. R3d). Detailed results and other analyses can be found in “Malignant cells in GC exhibit extensive heterogeneity” (Page 5-8, lines 89-142), “Identifying potential regulatory factors driving intestinal metaplasia” (Page 8-9, lines 144-165) in the revised manuscript and Supplementary Note 3 in the Supplementary Information (Page 32, lines 368-381).

Figure R3. Selected analyses of tumor cells about intestinal metaplasia and Wnt signaling.

(a) Violin plot showing the goblet scores (top) and enterocyte scores (bottom). **(b)** Heatmap showing the combined correlation of *CDX2*-associated genes by taking the product of correlation coefficients generated from scRNA-seq, in-house bulk RNA-seq as well as bulk RNA-seq from TCGA-STAD. Genes in red were predicted upstream-regulators of *CDX2*. **(c)** Dot plot showing the expression of Wnt-related genes. Dot size indicates proportion of expressing cells, colored by standardized expression levels. Genes in red indicated a low cellular detection rate outside epithelial cells. **(d)** Violin plot showing the expression of *CDX2*, *EPHB2* and *LGR5* in patients with or without APC mutation from TCGA-STAD dataset. * $p < 0.05$, ** $p < 0.01$, and *** $p < 0.001$ (Wilcoxon rank-sum test).

2. *The definition of tumor cells is problematic. In this study, tumor cells were defined solely based on CNV_score inferred from gene expression profiles; however, it is well known that a large fraction of primary gastric adenocarcinomas is genomically stable (GS) (Nature 2014, PMID: 25079317). Likewise, tissue enrichment analysis (Ro/e) can not be used to identify tumor cells, although theoretically, tumor tissues should contain more tumor cells than the adjacent normal tissues. However, due to sampling bias, tumor tissues may not always contain tumor cells and tumor cells can also found in adjacent normal tissues due to contamination or sampling issue. In addition, the definition of tumor-like clusters is confusing and a bit misleading. It is unclear how these tumor-like cells are defined. For example, cells of tumor-like cluster C4 mainly expressed normal epithelial lineage markers (PGA3, PGC, MUC6) and did not express marker genes for gastric cancer (Fig. 1i). Tumor-like cluster C13 mainly express marker gene of chief cells (PGA3, PGC) and parietal cells (ATP4A, ATP4B). The genes such as OLFM4, LGR5, and CD44, KRT7 can also express in normal epithelial cells. Clearly, the tumor like cells are transcriptionally heterogenous, as some cells are clustered closely with normal epithelial cells while some others are clustered closer to tumor cells. Analysis of tumor cells has to be significantly improved.*

Reply:

To make a more convincing definition of tumor cells, we combined multiple indicators from different aspects. Firstly, we compared the CNVs identified using WES data with the inferred CNVs from scRNA-seq (Fig. R4a). The results were consistent at the chromosome-level (Fig. R4b, Supplementary Fig. 2a and Supplementary Fig. 16) except for GC02. Secondly, we calculated tumor scores for each cell using expression patterns identified by Min Zhang *et al.*, 2021 (Fig. R4c, see Methods for details). Finally, we called tumor specific mutations for each patient by comparing the WES data of tumor tissue versus that of paratumor tissue and then, we searched the tumor specific mutations in the matching single-cell data. Such mutations were enriched in 4 clusters of epithelial cells (Fig. R4d). According to these results, as well as the expression patterns of cell type specific genes (Fig. R4e, Table R1), we finally defined four main cluster types, including

normal clusters, tumor clusters, intestinal metaplasia (IM) clusters, and an uncertain cluster (Fig. R4f). IM_enterocyte and IM_goblet were considered as precancerous clusters as they showed neither high inferCNV scores nor mutation enrichment of tumor specific mutations. The tissue enrichment level of each cluster of cells also agreed with this definition (Fig. R4g).

Table R1. Specific genes of epithelial cells

Cell types	Signature genes
chief cell	PGA4, PGA3, LIPF
gland mucous cell (GMC) (Mucous neck cell)	MUC6, FUT9
pit mucous cell (PMC) (Surface mucous cell)	MUC5AC, TFF1, TFF2, GKN1
parietal cell	ATP4A, ATP4B, GIF
goblet cell	MUC2, ATOH1, TFF3, SPINK4, CLCA1, FCGBP
enterocyte	FABP1, VIL1, CDX1, CDX2, REG4, KRT20

Figure R4. Distinguishing tumor cells and normal cells from multiple lines of evidences. (a) Inferred CNV profiles of epithelial cells of a representative patient (GC08) based on scRNA-seq dataset and whole-exome sequencing (WES) dataset. (b-d) UMAP of epithelial cells, colored by: (a) tumor score; (b) inferred CNV score; (c) the number of mutations. (e) Dot plot of marker genes for epithelial cell clusters. Dot size indicates proportion of expressing cells, colored by standardized expression levels. Genes in red color indicated a low cellular detection rate outside epithelial cells. (f) UMAP of epithelial cells. Clusters are labeled with inferred cell types. GMC, gland mucous cell; PMC, pit mucous cell; IM, intestinal metaplasia. (g) Tissue preference of each epithelial cluster estimated by Ro/e score.

(Please see the previous page for the legend)

3. *The analysis performed on tumor cells is very superficial. The authors simply defined the tumor cells clusters but without performed any further profiling on tumor cells. The insights derived from this part is very limited. Also, a clear limit is that, although ten tumors were sequenced, the epithelial and tumor cells were mainly derived from 4 tumors: GC08, GC10, GC07, GC02.*

Reply:

We thank the reviewer for point out the limitations of our analysis. We completely rewrote this part with an emphasis on the biological functions and clinical relevance of the findings. Given the limited number of single-cell samples, we combined bulk sequencing data and utilized expression and mutation information from TCGA-STAD to provide a secondary line of evidences. Detailed results and other analyses can be found in “Malignant cells in GC exhibit extensive heterogeneity” (Page 5-8, lines 89-142), “Identifying potential regulatory factors driving intestinal metaplasia” (Page 8-9, lines 144-165) in the revised manuscript and Supplementary Note 3 in the Supplementary Information (Page 32, lines 368-381).

4. *TASCs cells are interesting and the authors showed that the TASC signatures were associated with a worse survival in TCGA-STAD cohort. However, this survival difference may not be attributed to the presence of TASCs in the tumor microenvironment, because it is very possible that tumor cells and other cells may also express the signature genes of TASCs as the TASCs signature genes may not be exclusive to TASCs and the authors would need to provide more evidence on this. Also, potentially cofounding factors were not considered in their analysis. Similarly, DEGs of Macrophage_APOE could express in cells other than macrophages and TCGA data is not directly supportive of their biological significance and functional studies are needed.*

Reply:

Thank you for pointing out the potential confounding effect. We found that highly expressed genes were hardly cluster-specific (i.e., a gene is expressed by a single cluster in an exclusive fashion). To overcome this limitation, we applied MuSiC (Wang, X. et al., 2019), an algorithm to implement bulk tissue cell type deconvolution with scRNA-seq data. This

algorithm estimated proportions of different cell types in bulk RNA-seq data by a weighted non-negative least squares regression. After cell type deconvolution of bulk RNA-seq data from TCGA-STAD using MuSiC, we performed survival analysis with patients stratified according to the proportions of a certain cell type of interest. Our results showed that high proportions of Fib_1 and SMC_1 cells were significantly associated with worse patient survival. As for Macrophage_APOE and Endo_1, though both of them resulted in hazard ratios greater than 1, there was no statistical significance (Fig. R5).

Figure R5. The fraction of Fib_1 and SMC_1 is predictive of survival in the TCGA-STAD cohort. Kaplan-Meier curves of overall survival when stratifying the patients by high (top 40%) and low (bottom 40%) proportion of the respective cell type.

5. The TFs analysis in this study was very descriptive, the results were kind of isolated, and the significance of such analysis is not evident.

Reply:

We added experiment result showing the biological consequences of overexpressing the TFs we identified as important regulators of tumor associated macrophages. The experiment demonstrated that *APOE* and *APOC1* were upregulated in three different conditions after overexpressing *NR1H3* or *TFEC* in THP-1-derived macrophages (Fig. R6). As for stromal cells and tumor cells, we added discussion on the predicted TFs about the potential biological processes they involved in (angiogenesis and intestinal metaplasia, respectively). In the case of T cells, the TFs analysis was accompanied by pseudotime and trajectory analysis, providing insight into the potential regulatory TFs important for driving the cell differentiation trajectories. Besides, a list of predicted TF regulons that were enriched for specific clusters provided clues for further investigations.

Figure R6. Upregulation of APOE and APOC1 in overexpression experiments.

(a) Western Blot confirmed the overexpression of *NR1H3* and *TFEC* in THP-1 cells (b) *NR1H3* or *TFEC* overexpressed THP-1-derived macrophages were stimulated with lipopolysaccharide (LPS) + interferon γ (IFN γ) or Pam3CSK4. The expressions of *APOE* and *APOC1* were then measured by qPCR. Each column represents the mean \pm SEM of three independent experiments (n = 3). *p < 0.05, **p < 0.01, and ***p < 0.001 (Student's t test).

6. *The author stated that the DC_LAMP3 is less-described cluster, which is incorrect. This cell population has been well described by Dr. Zemin Zhang's group (Zhang et al. Cell 2019; Cheng et al. Cell 2021) and other groups. Also, the differentiation origin of LAMP3+ DC from cDC2 has been well described by Cheng et al. (Cell, 2021, PMID: 33545035), and therefore is not novel.*

Reply:

We agree with the reviewer and apologize for the misleading statements. We have now rephrased our statement in the revised manuscript (Page 13, lines 253-256).

7. *The authors identified 21 T cell clusters (Fig. 4a-b), some of these clusters are very small and not clearly separated from other cells. Also, the signature genes showed in 4b are largely shared among the CD4/CD8 cell subclusters. This raises concerns about the reproducibility of their clustering analysis and validations of the computational method are therefore warranted. It would be useful and necessary to demonstrate how robust the clustering is and the reproducibility of their results with additional clustering approaches (k-mean or hierarchical clustering based) and down sampling the cells (e.g. 25%, 50%).*

Reply:

Thank you for the suggestion. We down-sampled the T cells to 75%, 50%, 25% and 10%

of the cell population, separately. PCA was reproduced for these four subsets separately and then UMAP and four clustering approaches (Leiden, Louvain, k-means, and hierarchical clustering) were applied using the first 30 PCs (Fig. R7). The results showed that the clusters of the subsampled cells were close to the initial clustering result. .

To investigate the necessity of distinguishing some rare clusters, we looked for specifically expressed genes with clear biological significance. CD4_C1 and CD8_C1 were both naïve T cells expressing *LEF1*, *TCF7*, *CCR7*, and *SELL* (Fig. R7e), while they belonged to CD4⁺ T cells and CD8⁺ T cells respectively, which was an important difference. CD8_C6 and CD8_C7 were quite rare clusters, while the former showed distinctive expression of *ZNF683* (Tissue-resident T-cell transcription regulator protein) which indicated it was tissue-resident T cells. Besides, *GNLY* (Granulysin), an antimicrobial peptide, was also highly expressed by CD8_C6 but not by CD8_C7. As for CD8_C7, CD160 and KLRC1 were expressed at a much higher level. In the case of Treg_C1 and Treg_C3, the former expressed much higher levels of *LEF1*, *CCR7*, and *SELL*, indicating a naïve state, while the latter expressed higher levels of exhausted markers like *LAYN*. The list of cluster-specific genes can also be found in Fig. 5b.

In conclusion, the rare clusters had distinctively expressed genes with evidence of biological significance. Thus, it is necessary to distinguish these clusters instead of mixing them together.

Figure R7. Evaluation of the robustness and necessity of T-cell clusters

(a-d) Down-sampling of T cells to (a) 75%, (b) 50%, (c) 25%, and (d) 10% and clustering with different algorithms. (e) Dot plot showing selected markers to demonstrate the necessity of distinguishing these clusters: CD4_C1 and CD8_C1; CD8_C6 and CD8_C7; Treg_C1 and Treg_C3.

8. *The proportion of cells expressing the marker genes IL17A and IL23R in the two cell clusters CD4_C6_IL17A and CD8_C8_IL17A was very very low (Fig. 4b based on the size of the circles). This could possibly be due to the high gene dropout rate of 10x sequencing. Nevertheless, defining cells based on expression of these genes are not proper and has to be validated by additional approaches. The multicolor IHC showed in Fig. 4d is not convincing as only a several of such cells are shown in the images.*

Reply:

We apologize for only showing the IHC of a few cells in the previous version of the manuscript and we replaced it with better IHC results (Fig. R8a). As for the gene expression, we noticed that some cytokines were expressed in a low but extremely specific fashion. Low expressions resulted in more dropout events, thus it is unsuitable to display such cytokines together with other highly expressed genes in the same dotplot. Therefore, we showed the expression of *IL17A*, *IL17F*, *IL22*, *IL26*, and *IL23R* in a separated dotplot with a smaller maximum fraction value (Fig. R8b).

Figure R8. (a) multicolor IHC staining with anti-CD4, anti-CD8, and anti-IL17A antibodies, exemplified by patient GC988401 (View 1) and patient GC988419 (View 2 and View 3). The white and green arrows indicate $CD8^+IL17^+$ cells and $CD4^+IL17^+$ cells, respectively. The scale bar represents 20 μ m. **(b)** Dot plot showing interleukin-related genes expressed by Tc17 cells.

9. *Cell-cell communication analysis is potential interesting, but this part is purely descriptive without any functional evaluation. At least, correlation analysis should be performed to check whether the frequencies of those interacting cells are significantly correlated. The authors pooled cells from all samples and patients for Cell-cell communication analysis without take into consideration that these “interacting” cells (shown on the heatmap) may not even present in the same samples. For the methodology, it is unclear how the data are filtered, and whether those rare cell subsets with only few cells were excluded and the cutoff applied.*

Reply:

Thank you for the suggestion and for pointing out the pitfall. We reproduced the analysis separately for all the patients. In the revised manuscript, for a particular ligand-receptor pair, the percentage of patients with significant pair was shown by the color (Fig. R9a). The description of the methodology is revised. Besides, following your suggestion, we investigated the correlation between interacting cell types in our data as well as in TCGA-STAD data by applying MuSiC to estimate the proportions of those cell types. The result showed that Macrophage_APOE was positively correlated with TASCs (Fig. R9b-R9e).

Figure R9. (a) Dot plot of selected ligand–receptor interactions in tumors. Cell subsets are shown on the x axis, ligand (red) and receptor (black) pairs are shown on the y-axis. The color of circle denotes the proportion of patients with the significant interaction (p-value < 0.01) in the total patients with these interacting cell subsets. **(b-c)** Heatmap showing the spearman correlation between the proportions of different cell types in (b) TCGA-STAD dataset and (c) single-cell dataset. *p < 0.05, **p < 0.01, ***p < 0.001, and ****p < 0.0001. **(d-e)** Scatterplot showing the spearman correlation of cell proportions between Fib_1 and Mφ_APOE in (d) our single-cell dataset and (e) TCGA-STAD dataset.

10. Line 383-390: The authors described VDJ-gene usage, however, it is unclear how are the differences in VDJ-gene usages in T cell clusters related to the pathogenesis of gastric cancer.

Reply:

We apologize that we are not able to link the VDJ-gene usage to the pathogenesis of gastric cancer, given the limited number of patients. The original purpose of the VDJ-gene usage analysis was to indicate that T cells with different functions were likely to have different biases of VDJ-gene usages, instead of random usages. The underlying mechanism of this phenomenon is unclear and deserves further study.

11. Tc17 cells have to be functionally defined before further analyzing their developmental trajectories. What are the marker genes of these cells and what are their roles in anti-cancer immune response? T cell exhaustion trajectories have been well described in many other studies, and therefore it is unclear how results from current study can add to existing knowledge.

Reply:

We reorganized the paragraphs in the revised manuscript. Reported studies of Tc17 and our hypotheses for the functional role of Tc17 in gastric cancer were illustrated before the analysis of developmental trajectories. Marker genes of Tc17 were highly similar to that of Th17, but Tc17 also expressed $CD8^+$ T-cells-specific genes (Fig. R10a). Our results indicated that Tc17 cells may promote tumor progression through IL17, IL22, and IL26 signaling, however understanding the exact role of Tc17 in the TME requires further experiments. Clonotype and trajectory analysis suggested that Tc17 cells originated from tissue-resident memory T cells and could subsequently differentiate into exhausted T cells, providing an alternative pathway for T cell exhaustion distinct from the classic T cell exhaustion trajectories. The underlying mechanisms and functional significance between the two exhaustion trajectories warrant further investigation.

Figure R10. Marker genes of Tc17 were highly similar to that of Th17, but Tc17 also expressed *CD8⁺* T-cells-specific genes.

12. There is no description of doublets removal in the Methods, it is therefore not sure that doublets were carefully removed.

Reply:

The 12 main cell types in Fig. 1b were isolated and re-clustered separately. Subclusters expressing contradictory markers of known different cell types were removed as potential doublets. As suggested by reviewer 3, we listed the conventional markers of each cell type we used to identify the doublets in Table R2, accompanied by the number of excluded cells. In the case of T cells, we performed more stringent filtering: T cells with greater than 2 TRB or 2 TRA sequences were regarded as doublets. This resulted in a large number of excluded T cells. We updated the description in the Methods section (Page 32, lines 674-677).

Table R2. Specific genes of main cell types

Cell types	Signature genes	The number of excluded cells
B cells	MS4A1, CD19, BANK1, VPREB3, CD79A	1950
Plasma cells	JCHAIN, IGKC, MZB1, DERL3, CD79A	1066
Endocrine cells	CHGA, PCSK1N, TTR, SCG3, SCG5, EPCAM, KRT18, KRT19	152
Epithelial cells	MUC5AC, MUC1, S100P, LIPF, TFF1, TFF2, PSCA, EPCAM, KRT18, KRT19	3895
T & NK cells	CD2, CD3E, CD3D, CD3G, CD7	19407
Erythrocytes	HBB, HBA1, ALAS2, HBA2, CA1	0
Mast cells	TPSAB1, TPSB2, CPA3, TPSD1, GATA2	331
Myeloid cells	AIF1, CD68, CD14, FCN1, S100A9, MS4A7	1289
Endothelial cells	PLVAP, VWF, PECAM1, ACKR1, CLDN5, CD34	410
Fibroblasts	PDGFRA, PDPN, DCN, DPT, TRPA1	371
Smooth muscle cells	ACTA2, ACTG2, MYH11, RGS5, NDUFA4L2	197

13. *No statistical assessment of the batch effects is performed, an evaluation of the batch effects (whether it was present and how significant it was) were not described or in the Methods, or demonstrated in the main figures or supply figures.*

Reply:

We added a systematic evaluation of batch effects in Supplementary Note 1. Firstly, to evaluate the patient-specific features that might affect clustering, we investigated distributions of cell percentages contributed by different patients in each cluster (Fig. R11). Most of the clusters had diverse contributors roughly proportional to sample sizes (GC10 had the largest sample size), except for abnormal epithelial cells. Tumor cells were highly heterogeneous thus tumor cells from different patients formed separated clusters; Cells with transcriptional-level intestinal metaplasia (IM) were only detected in several patients, mainly GC08 and GC09, which was also confirmed by bulk RNA-seq (Fig. S2f). Cells in

epithelial clusters expressed highly specific marker genes (Fig. R4e, Table R1) thus the clustering was not dominated by unwanted batch effects. Besides, clusters consisting largely of cells from blood (e.g., Naïve T cells and Mono_CD14) had fewer contributors as the blood samples only came from GC06, GC07, GC08, GC09, and GC10. As for erythrocytes, very few of them passed through the ACK lysis buffer treatment.

We noticed that GC02 contributed 59% of the cells in CD8_C8_IL17A. We then plot the UMAP without GC02 (Fig. R11h) and found the shape of the cluster did not dramatically change. Besides, we counted the number of cells and the number of clonotypes contributed by each patient in the cell subset of trajectory analysis (Fig. R11i, R11j) (This subset consisted of cells that belonged to clonotypes detected in at least two clusters among CD8_C5_TOB1, CD8_C8_IL17A, and CD8_C9_HAVCR2). These demonstrated that result was not dominated by GC02 or other patient-specific features.

To evaluate the batch effect caused by cell quality, we first investigated the distributions of percentages of mitochondrial gene counts (abbreviated as “percent_mito” hereafter). The percent_mito raised when the cellular membrane was leaky or disrupted and cytoplasmic RNAs were released, thus it was considered as an indicator for cell quality. The percent_mito of samples from different patients were similar (Fig. R12h). We found significantly higher percent_mito in epithelial cells. This phenomenon was found in most of the samples hence it was not accidental, though the reason had not been figured out. A hypothesis was that epithelial cells from stomachs were more vulnerable to the experimental procedures than other cell types. The distributions of percent_mito varied in epithelial subclusters, but this didn’t seriously affect the clustering since each cluster expressed highly specific marker genes (Fig. R4e, Table R1). As for plasma cells (named Bcell_C7_SDC1 in B cells clustering), they had lower percent_mito due to the large number of total UMI counts, which was caused by a remarkable high expression of immunoglobulin-related genes. NKT_CD69 and Fib_3 showed higher percent_mito in NK cells and stromal cells, which might be affected by the cell-quality effect. Nevertheless, they would not affect the main results as our analyses did not focus on such cells.

The number of detected genes in a cell is another indicator of cell quality. A high

percent_mito combined with a low number of detected genes often indicates bad quality. However, biological differences can also influence the number of expressed genes. For example, the number of expressed genes is a robust indicator of developmental potential (Gulati et al., 2020). Besides, we found cycling cells had higher numbers of genes (Fig. R12i, R12l), which might be caused by the expression of cell-cycle genes. In addition, tumor cells seemed to express more genes than other cells (Fig. R12p).

In conclusion, the clustering was not obviously affected by patient-specific features. Though Fib_3 and NKT_CD69 seemed to be associated with low cell quality as they both had high percent_mito and low numbers of genes, we did not focus on them individually. Other clusters with unusual percent_mito or numbers of genes (e.g., tumor cells and cycling T cells) expressed highly specific marker genes. Therefore, we believed our main results were not significantly affected by batch effects.

Figure R11. (a-g) Stacked bar plot showing the patient distribution of each cluster. **(h)** UMAP plot showing the T cells without GC02. **(i-j)** Bar plots showing the number of clonotypes **(i)** and the number of cells **(j)** contributed by each patient in the cell subset of Tc17 trajectory analysis.

Figure. R12. (a-g) Violin plot showing the distribution of mitochondrial gene percentages in each cluster (h) Violin plot showing the distribution of mitochondrial gene percentages in each patient. (i) Scatter plot showing a positive correlation between the G2M score and the number of detected genes. Each dot represents a single cell. (j) Violin plot showing the distribution of the number of detected genes in each patient. (k-q) Violin plot showing the distribution of the number of detected genes in each cluster.

14. Potentially confounding factors were not considered in the survival analysis.

Reply:

Our understanding is that this problem is similar to that in comment 4. As described in the reply to comment 4, to control the potential confounding factors from other cell types, we applied MuSiC to bulk RNA-seq data from TCGA-STAD to make the survival analysis by stratifying the patients with inferred cell type proportions.

Minor comments:

15. Line: 141-143: it is difficult to read the two figure panels as the patient IDs were not labelled on Fig. 1i and the genes mentioned here (FABP1, MUC5AC, MUC5B) were not labelled on Fig. 1j.

Reply:

We apologize for the inconvenience. These figures are now replaced with new figures as we have rewritten this part completely.

16. Line 167: the author stated that Fib_1 expressed IL11 and IL24 while the expression levels of these two genes were very low and not easily visible in Fig. 2C.

Reply:

We noticed that it was not suitable to plot low expression genes and high expression genes together in the same dot plot. Thus, we plotted them separately with a smaller maximum fraction value (Fig. R13), and added this figure in Supplementary Fig. 3g.

Figure R13. Dot plot showing interleukin-related genes upregulated by Fib_1.

17. Figure 3b, the color key for standardized expression levels was not generated properly.

Reply:

We apologize for this mistake. We have updated this figure in Fig. 4b.

18. Line 206: no references added for “other studies”.

Reply:

We apologize for this mistake. We have added the references in the revised manuscript (Page 12, lines 226-233).

19. Figure 3c, the pathway name on the right bottom was not displayed properly.

Reply:

We apologize for this mistake. We have updated it in Fig. 4c.

20. It is not clear how the endocrine cells and smooth muscle cells were defined as no commonly used canonical markers of these cells are seen in their figure 1c or extended data figure 1d. The smooth muscle cells and fibroblasts had many overlapping genes and are not clearly distinguishable from each other on their figures.

Reply:

We apologize for this mistake. We have updated the dot plot (Fig. R14) with canonical markers in Table R2 and added this figure in Fig. 1d.

Figure R14. Signature genes for distinguishing different cell types.

21. Are GC03 and GC10 technical duplicates or biological duplicates? Were the libraries made using the same tissue?

Reply:

We apologize for this confusion. They were technical duplicates. The libraries were made using the same tissue. We added a detailed statement of the scRNA library preparation in the Methods section (Page 29, lines 599-601), and updated the Supplementary Fig. 1c.

Reference

Niu, B., Ye, K., Zhang, Q., Lu, C., Xie, M., McLellan, M. D., Wendl, M. C., & Ding, L. (2014). MSIsensor: microsatellite instability detection using paired tumor-normal sequence data. *Bioinformatics*, 30(7), 1015–1016.

Zhang, M., Hu, S., Min, M., Ni, Y., Lu, Z., Sun, X., Wu, J., Liu, B., Ying, X., & Liu, Y. (2021). Dissecting transcriptional heterogeneity in primary gastric adenocarcinoma by single cell RNA sequencing. *Gut*, 70(3), 464–475.

Wang, X., Park, J., Susztak, K., Zhang, N. R., & Li, M. (2019). Bulk tissue cell type deconvolution with multi-subject single-cell expression reference. *Nature communications*, 10(1), 380.

Gulati, G. S., et al. (2020). Single-cell transcriptional diversity is a hallmark of developmental potential. *Science*, 367(6476), 405–411.

Reviewer #3 (Remarks to the Author):

Sun et al. have performed a detailed analysis of cells infiltrating the gastric cancer (GC) tumor microenvironment, creating an atlas of >166,000 cells from 10 GC patients. Such cells were studied by scRNAseq, coupled with TCR sequencing. Among other findings, they report that tumor-associated macrophages and dendritic cells expressing LAMP3 were able to mediate T cell activity. Furthermore, they describe the role and function of IL-17 producing CD8+ T cells, a possible novel therapeutic target.

The paper is interesting and technically well done; the statistical analysis and the presentation of data are adequate.

Reply:

We really appreciate the reviewers' positive comments.

Comments:

1) Lines 603-605: "Cells expressing contradictory markers of known different cell types were removed as potential doublets". It would be better to indicate the markers used to identify the doublets and the number of excluded cells.

Reply:

Thank you for your suggestions. We have listed conventional markers of each cell type used to identify the doublets and the number of excluded cells in Supplementary Data File 2.xlsx. The contents are also present in Table R3 as follow. In the case of T cells, we performed more stringent filtering: T cells with greater than 2 TRB or 2 TRA sequences were regarded as doublets. This resulted in a large number of excluded T cells.

Table R3. Specific genes of main cell types

Cell types	Signature genes	The number of excluded cells
B cells	MS4A1, CD19, BANK1, VPREB3, CD79A	1950
Plasma cells	JCHAIN, IGKC, MZB1, DERL3, CD79A	1066
Endocrine cells	CHGA, PCSK1N, TTR, SCG3, SCG5, EPCAM, KRT18, KRT19	152
Epithelial cells	MUC5AC, MUC1, S100P, LIPF, TFF1, TFF2, PSCA, EPCAM, KRT18, KRT19	3895
T & NK cells	CD2, CD3E, CD3D, CD3G, CD7	19407
Erythrocytes	HBB, HBA1, ALAS2, HBA2, CA1	0
Mast cells	TPSAB1, TPSB2, CPA3, TPSD1, GATA2	331
Myeloid cells	AIF1, CD68, CD14, FCN1, S100A9, MS4A7	1289
Endothelial cells	PLVAP, VWF, PECAM1, ACKR1, CLDN5, CD34	410
Fibroblasts	PDGFRA, PDPN, DCN, DPT, TRPA1	371
Smooth muscle cells	ACTA2, ACTG2, MYH11, RGS5, NDUFA4L2	197

2) *Lines 625-629: "... SCTransform .. was used to remove a batch effect... ". The figures 4a and 4c report that some clusters are tissue-dependent, like CD8-C1, CD4-C1, CD4-C2. Naïve CD8 (CD8-C1) and naïve CD4 (CD4-C1) are close in the UMAP space, and it has to be excluded that was due to some source of technical variability. This variability can also cause the finding of other clusters. Have you tried to regress out other source of variations, like the mitochondrial gene percentages (that likely could be different among peripheral cells and TIL)? Additionally, some supplementary UMAP plot colored by the number of gene (and mitochondrial gene) expressed by each cluster could help in understanding the results of clustering process.*

Reply:

Thank you for your suggestions. We added a systematic evaluation of batch effects in Supplementary Note 1 accompanied by Fig. S13 and Fig. S14. Besides, an evaluation of

the robustness and necessity of T-cell clusters was added in Supplementary Note 2 accompanied by Fig. S15. In the case of T cells, the mitochondrial gene percentages and the number of genes of T cells are shown in Fig. R15 as follow. The distribution of mitochondrial gene percentages was balanced. Cycling T cells showed a high number of detected genes, which was likely caused by the expression of cell-cycle genes like MKI67.

For the example of Naïve CD8 (CD8-C1) and naïve CD4 (CD4-C1), they expressed the specific marker, *CD8* or *CD4*, in a mutually exclusive fashion, as we know that *CD8*⁺ T cells and *CD4*⁺ T cells are biologically different cell types.

Figure R15. Evaluation of batch effects in T cells. (a-b) Violin plot showing the mitochondrial gene percentages (a) and the number of detected genes (b) of T-cell clusters. (c-e) UMAP showing the mitochondrial gene percentages (c), the number of detected genes (d), and the expression of MKI67 (e) in T cells. (f) Dot plot showing specific marker genes of small T-cell clusters colored by standardized expression levels. Dot size indicates proportion of expressing cells.

3) *Figure 4: The clusters CD4_C2_LTB and CD4_C3_SLC2A3 designated as memory likely have to express also CREB or S100A4. If possible, highlining these transcripts would be indicative. Then, for the CD4_C4_CD69 cluster we recommend reporting the expression of CXCR6 and ANXA1, considering that cells were indicated as TRM (see: Szabo, Peter A., et al. "Single-cell transcriptomics of human T cells reveals tissue and activation signatures in health and disease." Nature Communications 10.1 (2019): 1-16).*

Reply:

Thank you for your suggestions. We found *CREB1* and *S100A4* were widely expressed by T cells (Fig. R16a). After meticulously reading this article (Szabo *et al.*, 2019), we found *ITGA1* was also a marker gene for TRM in addition to *CXCR6* and *ANXA1*. We have added *CREB1*, *S100A4*, *CXCR6*, *ANXA1* and *ITGA1* into Fig. S6d (the same as Fig. R16b as follow).

Figure R16. (a) Expressions of *CREB1*, *S100A4*, *CXCR6* and *ANXA1* of T cells on UMAP. **(b)** Dot plot showing the expressions of signature genes of *CD4*⁺ memory T cells colored by standardized expression levels. Dot size indicates proportion of expressing cells.

4) *Line 373: "We observed that CD8_C2(Effector), CD8_C3 (Cytotoxic), CD8_C4 (Effector memory), and CD8_C10 (MAIT) clusters have both higher proportions of clonal cells and proportions of clonal cells with shared TCR between blood and solid tissue (Fig. 6c)". Maybe using the percentage of overlapping TCR is not the optimal metrics to measure the shared clonotype among blood and solid tissue (or patient/cluster). The "Morisita similarity index", that also considers the size of the population, could be useful for this*

purpose.

Reply:

Thank you for your suggestion. We have calculated the “Morisita similarity index” and found the new result (Fig. R17a) is consistent in trend with our previous result (Fig. R17b), which does not affect our conclusion. We have changed the criteria of assessing the clonal migration between blood and tissues in the Methods section (Page 38-39, lines 810-817) and replaced the figure (Fig. 6c) with the new one.

Figure R17. The clonal migration between blood and tissues. (a) Comparison between proportions of clonal cells in each cluster (x axis) and Morisita-Horn similarity index calculated by TCRs shared across blood and solid tissues (y axis). **(b)** Comparison between proportions of clonal cells in each cluster (x axis) and percentage of cells with TCRs shared across blood and solid tissues (y axis).

5) *Line 398: “we thus hypothesized two possible trajectories for T cell state transition (Fig. 6i)”. This could be an interesting finding, but some additional plot showing the patient/sample effect would be indicative, and would greatly clarify if this observation is affected by the patient-effect rather than by the biological process.*

Reply:

Thank you for your suggestions. The patient-effect of the clustering was also evaluated in newly added Supplementary Note 1. In brief, we found GC02 contributed more than 50% of Tc17 cells. Therefore, we plotted the UMAP without GC02 (Fig. R18a) and found the shape of the cluster did not dramatically change. Besides, we counted the number of clonotypes and the number of cells contributed by each patient in the cell subset of Tc17 trajectory analysis (Fig. R18b and R18c) (This subset was used in Figure 6 for the velocity

analysis, which consisted of cells that belonged to clonotypes detected in at least two clusters among CD8_C5_TOB1, CD8_C8_IL17A, and CD8_C9_HAVCR2). In this subset, GC02 did not account for a large proportion. These demonstrated that the results were not dominated by GC02 or other patients alone.

Figure R18. Evaluation of the patient-effect in T cells. (a) Stacked bar plot showing the patient distribution of each T-cell cluster. (b) UMAP plot showing the T cells without GC02. (c-d) Bar plots showing the number of clonotypes (c) and the number of cells (d) contributed by each patient in the cell subset of Tc17 trajectory analysis.

Reference

Szabo, Peter A., et al. "Single-cell transcriptomics of human T cells reveals tissue and activation signatures in health and disease." *Nature Communications* 10.1 (2019): 1-16

REVIEWER COMMENTS

Reviewer #2 (Remarks to the Author):

The authors have addressed most of my comments and improved the manuscript.

1. The introduction and discussion should be updated to reflect the recent cancer discovery publication (Single-cell atlas of lineage states, tumor microenvironment and subtype-specific expression programs in gastric cancer. *Cancer Discov.* 2021 Oct 12:candisc.0683.2021. doi: 10.1158/2159-8290.CD-21-0683) and could be framed more tightly as to what questions remain after this work and how their study is different.
2. Given the recent cancer discovery publication, this study is descriptive. If it is feasible, I strongly suggest functionally validating one of the most interesting novel discoveries they made.
2. Figure R7. panels a-d, it is hard to just visualize the differences between UMAP plots, I would suggest the authors quantify them using other suitable approaches, such as correlation analysis. also, it is not clear whether the number of clusters is reproducible, are there any clusters to be merged or split based on the updated analysis?
3. only a subset of the reported epithelial cell lineages are identified, some others such as basal gland cells, tuft cells, or other reported progenitor cells were not described.
4. the data they deposited is not accessible, would suggest depositing their data to other commonly used databases such as GEO and EGA.

Point-by-point responses to the reviewers' comments

Reviewer #2 (Remarks to the Author):

The authors have addressed most of my comments and improved the manuscript.

Reply:

Thank you for your comments and criticisms.

Comment 1: The introduction and discussion should be updated to reflect the recent cancer discovery publication (Single-cell atlas of lineage states, tumor microenvironment and subtype-specific expression programs in gastric cancer. Cancer Discov. 2021 Oct 12: candisc.0683.2021. doi: 10.1158/2159-8290.CD-21-0683) and could be framed more tightly as to what questions remain after this work and how their study is different.

Reply:

Thank you for providing the information about this recent publication. We appreciated the discoveries and validations in this work. We integrated their results into the introduction section as “Kumar et al. revealed an increased plasma cell proportions in diffuse-type gastric tumors and studied the INHBA-FAP axis in cancer-associated fibroblasts.” (Page 3, lines 54-55) and into the result section as “Kumar et al. reported that recombinant INHBA was sufficient to upregulate the expression of FAP and collagen genes in normal fibroblast lines.” (Page 11, lines 219-221).

Comment 2: Given the recent cancer discovery publication, this study is descriptive. If it is feasible, I strongly suggest functionally validating one of the most interesting novel discoveries they made.

Reply:

Thank you for your suggestions. We carried out several experiments to validate some of our analysis results.

As we predicted several transcription factors (TFs) that might regulate *CDX2*, we conducted overexpression experiments to validate if there was any regulation between *CDX2* and these TFs. We collected several gastric cancer cell lines and measured their expression of *CDX2* (Fig. R1a). We first overexpressed *HOXA13* or *NR112* in SGC-7901 and MKN-28 cell lines (Fig. R2b). The western blot result showed that *HOXA13* could upregulate *CDX2* in SGC-7901 cell lines (Fig. R2c). Then, we overexpressed *CDX2* in SGC-7901, MKN-28 and HGC-27 cell lines (Fig. R2d). We found that the overexpression of *CDX2* had an influence on the morphology of these cell lines, making them elongated (Fig. R2e). The delivery times of rarely used antibodies were too long due to the COVID-19 pandemic, thus we only validated the expression of these TFs with qPCR (Fig. R2f). The result showed varying degrees of upregulation of *HNF4A*, *HOXA13*, *NR5A2*, *NR112* and *CDX1* in cell lines with *CDX2* overexpression. According to these results, *HOXA13* and *CDX2* might form a positive feedback loop under certain circumstances. The underlying molecular mechanism deserves further study.

Cell-cell communication analysis revealed that fibroblasts in tumors could support tumor growth by secreting *HGF*, *FGF7*, *BDNF* and other cytokines, and promote the polarization of macrophages by secreting *CSF1* and *IL34*. Here, we carried out *in vitro* experiments with gastric cancer-associated fibroblasts (CAFs) (Fig. R2). We found that CAFs-derived supernatants had the capacity to support tumor growth for several gastric cancer cell lines (Fig. R2b). Moreover, *in vitro*

co-culture experiment evinced that certain cytokines secreted by CAFs could induce THP-1 monocyte-derived macrophages into the states of macrophages in the GC TME via upregulating the marker genes of both M ϕ _APOE and M ϕ _THBS1 (Fig. R2c). The M ϕ _APOE score showed a continuous increase, while the M ϕ _THBS1 score displayed slight variation during the induction (Fig. R2d). We also found cytokines predicted to interact with tumor-associated stromal cells (TASCs), such as *IL6* and *OSM*, were highly expressed by induced macrophages (Fig. R2e).

Figure R1. The effect of CDX2 overexpression in gastric cancer cell lines. (a) Baseline expression of nine gastric cancer cell lines, including SNU-16, KATO III, MKN-45, SNU-1, AGS, SGC-7901, HGC-27, MKN-28 and MKN-7, were measured by western blot. (b) Western blot analysis to confirm the overexpression of HOXA13 and NR1I2 in SGC-7901 and MKN-28 infected with negative control (ctrl), HOXA13, or NR1I2 overexpression lentivirus. (c) Western blot analysis to detect the expression of CDX2 in SGC-7901 and MKN-28 infected with negative control (ctrl), HOXA13, or NR1I2 overexpression lentivirus. (d) Western blot analysis to confirm the overexpression of CDX2 in SGC-7901, MKN-28 and HGC-27 infected with negative control (ctrl) or CDX2 overexpression lentivirus. WT, wildtype cell line. (e) Effect of CDX2 overexpression on the morphology of SGC-7901, HGC-27, and MKN-28. (f) The expressions of HNF4A, HOXA13, NR5A2, NR1I2 and CDX1 were measured in SGC-7901, MKN-28 and HGC-27 infected with negative control (ctrl) and CDX2 overexpression lentivirus by qPCR. Each column represents the mean \pm SD of three independent experiments (n = 3). *p < 0.05, **p < 0.01, and ***p < 0.001 (Student's t-test).

Figure R2. The effect of gastric cancer-associated fibroblasts on the proliferation of tumor cells and the polarization of macrophages. (a) Mesenchymal marker vimentin expression in gastric cancer-derived fibroblasts was detected by immunofluorescence (100 X). **(b)** The supernatants from GC CAFs were collected by centrifugation as conditioned medium (CM), and incubated with six GC cell lines for 60 h, including MKN-7, AGS, SGC-7901, HGC-27, SNU-1 and MKN-28. Cell survival was determined by CCK-8 assay. **(c)** Heatmap showing the mean expression of the marker genes of both Mφ_APOE and Mφ_THBS1 in THP-1 monocyte-derived macrophages that were co-cultured with gastric CAFs in a transwell system, cultured by gastric CAFs conditioned mediums (CM), or cultured by standard medium for 24 h, 48 h, 60 h, 72 h. **(d)** Bar plot showing the Mφ_APOE and Mφ_THBS1 scores. Each column represents the mean \pm SD of three duplicates. **(e)** Heatmap showing the mean expression of cytokines predicted to interact with TASCs.

Comment 3: Figure R7. panels a-d, it is hard to just visualize the differences between UMAP plots, I would suggest the authors quantify them using other suitable approaches, such as correlation analysis. also, it is not clear whether the number of clusters is reproducible, are there any clusters to be merged or split based on the updated analysis?

Reply: Thank you for the suggestion. We calculated normalized mutual information to evaluate the similarity between different clustering results (Fig. R3a-e). The results of Leiden and Louvain were similar to each other. These two clustering methods were preferred by several widely used integrated tools such as Seurat⁽¹⁾ and Scanpy⁽²⁾ and gained better overall performance on clustering tasks for single-cell RNA sequencing data compared to hierarchical clustering and k-means, according to a recent multiple-criteria benchmarking research⁽³⁾.

With regard to the number of clusters, it depends on an adjustable hyperparameter inside each clustering method, thus similar numbers of clusters can always be reproduced by different methods according to the desired resolution of clustering. The merging and splitting of clusters by different methods at similar resolution were of great importance indeed. In brief, there was no obvious split cluster in these analyses, while there were 3 cases of merging: CD4_C1 and CD8_C1; CD8_C6 and CD8_C7; Treg_C1 and Treg_C3. We carefully analyzed the difference between merged clusters and found biologically meaningful differences in gene expression (Fig. R3f).

CD4_C1 and CD8_C1 were merged in the results of hierarchical clustering and k-means in all sample sizes and merged by Louvain when sub-sampling 25% and 10% of the cells. These two clusters were both naïve T cells expressing SELL, LEF1 and TCF7, while they belonged to CD4⁺ T cells and CD8⁺ T cells respectively according to the expression of CD4, CD8A and CD8B. Therefore, CD4_C1 and CD8_C1 should not be merged.

CD8_C6 and CD8_C7 were merged frequently. The former expressed ZNF683 (Tissue-resident T-cell transcription regulator protein) and GNLY (Granulysin), an antimicrobial peptide, while the latter expressed CD160 and KLRC1. Thus, we suggested CD4_C1 and CD8_C1 should not be merged.

As for Treg_C1 and Treg_C3, the former expressed much higher levels of LEF1, CCR7, and SELL, indicating a naïve state, while the latter expressed higher levels of exhausted markers like LAYN. Hence, we suggested Treg_C1 and Treg_C3 should not be merged.

In conclusion, though some clusters were merged in some situations, we suggested they should not be merged considering the differences of gene expression. Besides, our analyses did not focus on these cell clusters, thus our results would not be influenced even if these clusters were merged.

Figure R3. (a-e) Heatmaps showing normalized mutual information evaluating the similarity between different clustering results at different subsampling rate. (f) Differentially expressed genes of clusters which were merged in some clustering situations.

Comment 4: only a subset of the reported epithelial cell lineages are identified, some others such as basal gland cells, tuft cells, or other reported progenitor cells were not described.

Reply: We apologized for using gland mucous cell (GMC) as an unclear name. Actually, these cells indicated basal gland cells. We update the name as basal gland mucous cells (GMC) in the revised manuscript.

For progenitor cells, they are rare and require a large amount of epithelial cells to be clearly distinguishable. When there is only a small amount of progenitor cells, they are likely to be clustered together with other cells.

As for tuft cells, DCLK1 was reported as the most widely used marker gene⁽⁴⁾, but we found very rare expression of DCLK1 in our data (Fig. R4a). However, some other reported marker genes such as SH2D6, AVIL, and TRPM5A were expressed by a small bunch of cells closed to goblet cells in our data (Fig. R3b-h). We speculated that these cells were tuft cells, while the number of these cells was so small that clustering algorithms could not isolate them.

Figure R4. UMAPs showing the expression of reported marker genes of tuft cells in epithelial cells.

Comment 5: the data they deposited is not accessible, would suggest depositing their data to other commonly used databases such as GEO and EGA.

Reply: We apologized that the processed data was not appropriately deposited before. Now, the processed expression matrices and cell annotations have been deposited into Open Archive for Miscellaneous Data (OMIX) database with accession ID: OMIX001073 and can be previewed at: <https://ngdc.cncb.ac.cn/omix/preview/UleSVabl>.

The raw data of this research have been deposited into Genome Sequence Archive (GSA) for human with accession ID: HRA000704 and can be previewed at: <https://ngdc.cncb.ac.cn/gsa-human/s/rDZ3PhnM>.

These preview links only provide file lists of data. The release date will be set to the publication date in the future. Once released, the processed data will be available at: <https://ngdc.cncb.ac.cn/omix/release/OMIX001073> and the raw data will be available at: <https://ngdc.cncb.ac.cn/gsa-human/browse/HRA000704> (unaccessible now).

According to Decree No. 717 of the State Council of the People's Republic of China, to manage and protect Chinese human genetics resources, the sharing of human data must be compliant with the "Guidance of the Ministry of Science and Technology (MOST) for the Review and Approval of Human Genetic Resources". Therefore, we are encouraged to deposit data using databases developed by the National Genomics Data Center (NGDC) of China National Center for Bioinformatics (CNCB), including OMIX database and GSA database, which are directly linked to the registration and backup systems of MOST.

Reference

1. Stuart T, Butler A, Hoffman P, Hafemeister C, Papalexi E, Mauck WM, et al. Comprehensive Integration of Single-Cell Data. *Cell*. 2019;177(7):1888-902.e21.
2. Wolf FA, Angerer P, Theis FJ. SCANPY: large-scale single-cell gene expression data analysis. *Genome Biology*. 2018;19(1).

3. Yu L, Cao Y, Yang JYH, Yang P. Benchmarking clustering algorithms on estimating the number of cell types from single-cell RNA-sequencing data. *Genome Biology*. 2022;23(1).
4. Ting H-A, von Moltke J. The Immune Function of Tuft Cells at Gut Mucosal Surfaces and Beyond. *The Journal of Immunology*. 2019;202(5):1321-9.

Reviewer #3 (Remarks to the Author):

This referee is happy with the changes and has no specific comments to the revised version.

Reply: We are glad and grateful for your affirmation and mediation.

REVIEWERS' COMMENTS

Reviewer #2 (Remarks to the Author):

The authors have successfully addressed most of my comments and further improved the manuscript. I'm satisfied with the revised manuscript and have no additional comments.

Reviewer #2 (Remarks to the Author):

The authors have successfully addressed most of my comments and further improved the manuscript. I'm satisfied with the revised manuscript and have no additional comments.

Reply: Thank you very much for your valuable comments.